# DNA Methylation as an Early Indicator of Aging in Stored Seeds of “Exceptional” Species *Populus nigra* L.

**DOI:** 10.3390/cells11132080

**Published:** 2022-06-30

**Authors:** Marcin Michalak, Beata Patrycja Plitta-Michalak, Mirosława Zofia Naskręt-Barciszewska, Jan Barciszewski, Paweł Chmielarz

**Affiliations:** 1Department of Plant Physiology, Genetics and Biotechnology, University of Warmia and Mazury in Olsztyn, M Oczapowskiego 1A, 10-721 Olsztyn, Poland; beata.plitta-michalak@uwm.edu.pl; 2Department of Chemistry, University of Warmia and Mazury in Olsztyn, Plac Łódzki 4, 10-719 Olsztyn, Poland; 3Institute of Bioorganic Chemistry, Polish Academy of Sciences, Z. Noskowskiego 12/14, 61-704 Poznan, Poland; miroslawa.barciszewska@ibch.poznan.pl (M.Z.N.-B.); jan.barciszewski@ibch.poznan.pl (J.B.); 4NanoBioMedical Centre, Adam Mickiewicz University, Wszechnicy Piastowskiej 3, 61-614 Poznan, Poland; 5Institute of Dendrology, Polish Academy of Sciences, Parkowa 5, 62-035 Kornik, Poland; pach@man.poznan.pl

**Keywords:** black poplar, DNA methylation, epigenetics, gene banks, long-term storage, seeds, seed aging, short-lived seeds, viability loss, 5-methylcytosine

## Abstract

Ex situ preservation of genetic resources is an essential strategy for the conservation of plant biodiversity. In this regard, seed storage is the most convenient and efficient way of preserving germplasm for future plant breeding efforts. A better understanding of the molecular changes that occur during seed desiccation and aging is necessary to improve conservation protocols, as well as real-time methods for monitoring seed quality. In the present study, we assessed changes in the level of genomic 5-methylcytosine (5mC) in seeds of *Populus nigra* L. by 2D-TLC. Epigenetic changes were characterized in response to several seed storage regimes. Our results demonstrate that *P. nigra* seeds represent an intermediate type of post-harvest behavior, falling between recalcitrant and orthodox seeds. This was also true for the epigenetic response of *P. nigra* seeds to external factors. A crucial question is whether aging in seeds is initiated by a decline in the level of 5mC, or if epigenetic changes induce a process that leads to deterioration. In our study, we demonstrate for the first time that 5mC levels decrease during storage and that the decline can be detected before any changes in seed germination are evident. Once *P. nigra* seeds reached an 8–10% reduction in the level of 5mC, a substantial decrease in germination occurred. The decline in the level of 5mC appears to be a critical parameter underlying the rapid deterioration of intermediate seeds. Thus, the measurement of 5mC can be a fast, real-time method for assessing asymptomatic aging in stored seeds.

## 1. Introduction

The most efficient and effective method of ex situ, long-term conservation of plant diversity is the storage of seeds in seed banks. Currently, there are more than 1750 seed banks worldwide. For many decades, these facilities were focused on the conservation of food crop species; however, their scope has now widely broadened to include crop-related wild species and endangered plant species in general. In fact, many thousands of seed samples from wild species have been placed into long-term storage since the Global Strategy for Plant Conservation (GSPC) was adopted in 2002 [1]. It is expected that seed bank collections of wild species will play an increasingly important role in habitat restoration and the reintroduction of species that have been lost due to development and climate change [2]. These facts underline the importance of developing effective methods of seed collections that ensure their long-term viability for future use.

Unfortunately, the goal of the GSPC to preserve “at least 75% of threatened plant species in ex situ collections by 2020” [3] has not been achieved in most countries [4]. One of the main reasons underlying this shortfall is that seed banking is based on desiccating seeds at 10–25% relative humidity (RH) to a specific moisture content (MC) followed by storage at −20 °C. This protocol is efficient for long-term conservation of orthodox seeds [5,6]. Unfortunately, many other species can be described as “exceptional” which means that their seeds cannot be stored under conventional seed bank conditions. It is estimated that one-third of all exceptional species globally are threatened with extinction [7,8]. Four main requirements need to be met for successful seed banking, and species designated as exceptional are placed into different groups based on which of the four Exceptional Factors (EF) is the most predominant [7]. The second largest EF group (EF3) of known exceptional species (36.6%) are characterized as species producing short-lived seeds [8]. Fundamental knowledge about the progressive loss of seed viability during storage is based on studies conducted in crops and model species (*Arabidopsis thaliana* L.) which produce orthodox seeds with a relatively long lifespan [9,10,11,12]. In the case of short-lived seeds, however, our knowledge of the factors determining seed viability is insufficient, which prevents the establishment of effective methods for long-term storage. In this respect, it has recently been suggested that aging mechanisms may be different in seeds of different species based on their cell’s “dry architecture”, particularly in relation to seeds with a short lifespan [13]. For example, mature *Populus nigra* L. seeds have low amount of storage lipids and contain developed chloroplasts which may be the main source of oxidative stress inducing the fast aging in these seeds [13,14]. On the other hand, *A. thaliana* seeds do not contain chloroplasts but have relatively high amounts of storage lipids that, instead of the chloroplast, may be involved in the slower aging of the seeds in this and other oily species [13,15,16].

Despite potential differences in aging mechanisms between *P. nigra* and *A. thaliana*, black poplar seeds have been chosen for this research as this species is one of the most endangered tree species in Europe [17]. In Poland, this species is not legally protected as a species, but most populations are part of the Natura 2000 conservation program that prevents devastation of ecologically valuable ecosystems. However, a progressive reduction in black poplar population size is still being observed in Polish river landscapes [18]. Restoration and protection of riparian forest species are a key priority in biodiversity conservation and climate change strategies. Therefore, there is an interest in protecting and preserving *P. nigra* germplasm as it is a pioneer species and a key component of softwood forests in Europe [19,20]. According to the Global List of Exceptional Plants (https://cincinnatizoo.org/global-list-of-threatened-exceptional-plants/, which was accessed on 7 February 2022), *P. nigra* is classified as an exceptional species (EF3) producing short-lived seeds. An alternative classification of *P. nigra* seeds based on post-harvest storage response characterizes them as intermediate seeds [21] and places them between orthodox (desiccation tolerant) and recalcitrant (desiccation sensitive) seeds. Black poplar seeds rapidly lose their viability in storage [22,23], although it is possible to extend the span of their seed viability up to 1–2 years [24,25]. In relation to aging, the physiological and biochemical changes in *P. nigra* seeds have been the subject of several studies [17,23,24,25,26,27,28]. Specifically, these studies have demonstrated that storage of black poplar seeds results in increased oxidative stress [27], large changes in protein abundance [28], and a reduced proportion of roots with absorptive function [26]. To reiterate, many of these changes could be induced by an exacerbated oxidative stress mediated by the chloroplasts found in the mature seeds of this species [13,14]. None of the documented changes, however, can be used as a marker of seed viability as no clear correlation has been established between their viability and changes in any of the measured parameters.

DNA methylation is a conserved, epigenetic modification that regulates gene expression and affects genome stability. Epigenetic factors play a major role in plant development and environmental stress response, especially in species with a complex genome [29]. Therefore, there is a need to assess the role of DNA methylation in tree species that have large genomes. In particular, the effect of seed aging on DNA methylation integrity was addressed in several previous research studies, showing increased epigenetic instability in stored *Secale cereale* L. and *Mentha aquatica* L. seeds as well as in seedlings [30,31]. However, in that research accelerated, aging protocols (a high moisture content of seeds and a high temperature of storage) were used to speed up the aging process. Therefore, observed changes may not be comparable to changes observed during slow aging (in seed banks), as seed deterioration processes may occur through different mechanisms, depending on temperature conditions and low or high MC of seeds due to differences in molecular mobility and biochemical reaction rates within cytoplasm [13,16,32,33]. Age-related changes in the global 5mC level were also observed in *Quercus robur* L. acorns. However, these seeds belong to the recalcitrant category; therefore, they were not desiccated prior to storage and remained at a high MC of 38% [34]. Therefore, the choice of *P. nigra* seeds has given us the opportunity to investigate the changes in DNA methylation in aging seeds over a short time at storage conditions relevant to seed banking (dry and cold).

The objective of the present research was to determine if changes in global 5mC levels are correlated with the viability of short-lived, intermediate seeds produced by exceptional species. Seeds were kept moist or dried at various temperatures to observe a viability decline and test the hypothesis that measurements of changes in the global percentage of 5mC can serve as an early indicator of a seed viability reduction regardless of their initial viability and moisture content.

## 2. Materials and Methods

### 2.1. Plant Material and Assessment of Moisture Content (MC)

Mature seeds of black poplar (*Populus nigra* L.) were collected from three different 50- to 70-year-old trees in one population growing near Czeszewo (52°8′ N, 17°30′ E) that were physically separated from each other by a distance of at least 3 km. Seeds were collected in the same manner as described in [25]. No other poplar plantations or poplar species were located nearby, ensuring that the seeds used in the study were true seeds of *P. nigra*.

Seed MC (3 replicates of 50 seeds each) was determined by drying seeds at 103 ± 2 °C for 17 h. Freshly collected seeds exhibit an MC ranging from 12.5–14.5%. The harvested seeds were subsequently dried in the laboratory to a range of 7.6 to 7.8% (Table 1). Seeds were then stored for one year at a series of different temperatures: +10 °C, +3 °C, −20 °C, and −196 °C in liquid nitrogen as described [24].

### 2.2. Germination

Germination tests were conducted using 50 seeds placed on moist filter paper (70 mm in diameter) in a Jacobsen apparatus type 5101 (Rumed, Laatzen, Germany) covered with a plastic lid. The 50 seeds served as one biological replicate, and germination in a total of four replicates was assessed for each level of desiccation and storage temperature. Temperature was maintained at 23 °C for 22 h and 27 °C for 2 h each day, and light was provided on a 12 h cycle (irradiance of 22 μmol m^−2^ s^−1^) with illumination being provided during the period with the highest daily temperature. All seeds with an emerging radical, collet hairs, and fully expanded cotyledons after 14 days were considered as germinated.

### 2.3. DNA Isolation and Assessment of Global DNA Methylation Levels

A TLC-based method was used to measure the level of global DNA methylation. Total genomic DNA was extracted with a Qiagen DNeasy Plant Mini Kit^TM^ (Qiagen, Hilden, Germany). Each biological replicate was comprised of 0.1 g of seeds, corresponding to approximately 100 seeds, and 4 replicates were assessed in each treatment. Procedure-based variability in 5mC measurements were minimized by repeating each assessment five times.

The analysis of the global level of 5mC in DNA in seeds was conducted and calculated as previously described [35,36,37,38]. High purity dried DNA (1 µg) (A_260_/_280_ approx. 1.8) was digested to completion with 0.001 U of spleen phosphodiesterase II and 0.02 U of microccocal nuclease in a 20 mM succinate buffer containing 10 mM CaCl_2_ for 6 h at 37 °C. The resulting hydrolysate (0.3 µg) was labelled with 1.6 µCi (γ^32^P) ATP (6000 Ci/mmol Hartmann Analytic, Braunschweig, Germany) and 1.5 U of T4 polynucleotide kinase in 10 mM bicine-NaOH buffer (pH 9.7) containing 10 mM MgCl_2_, 10 mM DTT and 1 mM spermidine. After incubation for 30 min at 37 °C, 0.03 U of apyrase in 10 mM bicine-NaOH buffer was added, and the mixture was incubated for 30 min. Subsequently, 0.2 µg of RNase P1 in 500 mM ammonium acetate buffer (pH 4.5) was used for phosphate cleavage. The analysis of (γ^32^P) 5mC was performed by 2-dimensional thin layer chromatography (2D TLC) on cellulose plates (Merck, Darmstadt, Germany) in isobutyric acid/NH_4_OH/H_2_O (66 mL/1 mL/17 mL, *v*/*v*/*v*) (first direction), and 0.1 M sodium phosphate pH 6.8/ammonium sulphate/n-propanol (100 mL/60 g/1.5 mL, *v*/*w*/*v*) (second direction). Radioactivity was measured with a Fluoro Image Analyzer FLA-5100 and Multi Gauge 3.0 software (Fuji Photo Film Co., Ltd. Tokyo, Japan).

For quantitative determination of 5mC content, fluoroscopic image analysis was used to determine the content of cytosine (C) and 5mC. All calculations of global DNA methylation levels were made based on measurements of the intensity of individual spots corresponding to the analyzed nucleotide, using the following formula:R (%) = 5mC/total C (5mC and C) × 100(1)

### 2.4. Statistical Analysis

R software version 4.1.0 (www.r-project.org) was used for the statistical analyses and graphical visualization of data. The effect of desiccation and temperature on seed germination was separately evaluated using a generalized linear model (GLM) with a binomial distribution and Logit link function [39,40,41]. The impact of desiccation and temperature on the level of DNA methylation was evaluated using a linear model. The assessment of normality of the 5mC data was conducted using the Shapiro–Wilk test. A Levene’s test was used to test the homogeneity of variance across groups.

The Wilcoxon–Mann–Whitney test was conducted to determine significant differences in germination or DNA methylation in response to desiccation on each of the three seed lots individually.

A one-way ANOVA was used to determine significant differences between mean values during the storage of seeds. Pairwise comparisons between treatments were performed using a Duncan’s multiple range test at *p* ≤ 0.05. In the case of ANOVA conducted on GLM, *p*-values were calculated using a chi-squared test and chi-squared distribution. The relationship between germination and DNA methylation was determined using a Pearson correlation coefficient analysis. The analysis of the effect of seed storage on germination and DNA methylation level was conducted separately for each seed lot and MC, as well as for the combined data.

The prediction of relative change (Δ%) in 5mC percentage in *P. nigra* seeds leading to initiation of seed viability decline during storage was modeled using the glm function with a binomial distribution available in R. The GLM model was constructed on all germination and all global DNA methylation level data. The level of 5mC at 90%, 80%, 50%, 25%, 10%, and 5% of germination was calculated using the dose.p function on the basis of constructed models. The significance of the models is provided in Appendix A. As a control, to calculate relative changes in the DNA methylation level, data obtained from the model at 95% of germination was used as the dose.p function does not allow calculation values close to 0% or 100% with high accuracy, and the germination median value for non-stored *P. nigra* seed was close to 95%. To construct the graph, the relative change (Δ%) in percentage between the percentage of 5mC calculated for germination at 95% and the percentage of 5mC at an arbitrary chosen germination level was calculated. The results and detailed calculation of Δ% in 5mC are provided in Appendix A. The same approach was used to construct the graph for all seeds, seeds at ~12–14% of MC, and desiccated seeds at 7% of MC. The R-package ‘ggplot2’ was used for graphical visualization of the data.

## 3. Results

### 3.1. Seed Germination following Desiccation

The impact of desiccation on the germination of *P. nigra* seeds obtained from three different seed lots was assessed. Results indicated that there were no significant differences between non-desiccated and desiccated seeds (Figure 1). Non-desiccated seeds from Seed Lot 1 did, however, exhibit a lower initial seed germination capacity in comparison to other seed lots, thereby facilitating the study of molecular marker changes in seeds of varied initial viability (Figure 1).

### 3.2. DNA Methylation Level following Seed Desiccation

Freshly collected seeds from Seed Lot 1 were characterized by a 5mC level equal to R = 6.96%. Desiccation of those seeds to an MC of 7.7% resulted in a significant increase in the 5mC level (R1 = 8.48%), (Figure 2). Similar changes in 5mC levels in response to desiccation were also observed in Seed Lots 2 and 3, which exhibited an increase from 8.38% to 10.1% and 8.75% to 9.74%, respectively, (Figure 2).

### 3.3. Seed Germination after One Year of Storage at Different Moisture Contents and Temperatures

Cryogenic storage of Seed Lot 1 at a 14.5% MC exhibited the highest germination capacity (93%). A significantly lower germination (64.2%) was observed for seeds from the same seed lot that were stored at −20 °C. Seeds from the same seed lot exhibited a complete loss of germination capacity when stored at +3 and +10 °C (Figure 3a). A lower MC value, however, did improve germination of those seeds, especially in seeds stored at +3 °C (Figure 3b). Little improvement, if any, was observed at +10 °C.

Seeds at a 13.6% MC from Seed Lot 2 exhibited a germination capacity of 93.8%. A similar level of germination was observed after one year of storage under cryogenic conditions (Figure 3c). One-year storage at −20 °C resulted in a decrease in the germination to 64.2%, while storage of the same seeds at higher temperatures completely abolished the germination capacity (Figure 3c). Desiccation of those seeds to a 7.8% MC improved the total germination capacity of seeds stored at +10 °C, +3 °C, −20 °C (Figure 3d).

Seeds from Seed Lot 3 at a 12.5% MC exhibited the highest rate of germination (99.5%) after one year of storage at both −196 °C and −20 °C, a level of germination that was similar to non-stored seeds. Seeds with a 12.5% MC that were stored at +3 °C or +10 °C, however, exhibited a complete loss in germination capacity (Figure 3e). Desiccation of those seeds to a 7.6% MC improved the germination of seeds stored at +10 °C or +3 °C (Figure 3f).

### 3.4. DNA Methylation Level in Seeds after One Year of Storage at Different Moisture Contents and Temperatures

The level of global DNA methylation in non-stored non-desiccated seeds of Seed Lot 1 at a 14.5% MC was R = 6.96%. A similar level of 5mC (R = 7.19%) was observed in seeds with the same MC stored for one year at −20 °C. Significantly lower global levels of 5mC were observed in seeds stored for one year at +3 °C or 10 °C, R = 5.73% and R = 5.69%, respectively. Seeds stored under cryogenic conditions had the highest level of 5mC (R = 8.10%) (Figure 4a). Desiccation resulted in non-significant differences in 5mC levels in seeds stored at low temperatures of −20 °C and −196 °C. Significantly lower levels of 5mC were observed in seeds stored at temperatures of +3 °C and +10 °C, R = 6.58% and R = 6.05%, respectively (Figure 4b).

Non-stored non-desiccated *P. nigra* Seed Lot 2 at a 13.6% MC had a 5mC level of R = 8.38%, a level that was similar to seeds stored under cryogenic conditions (R = 7.90%). A decrease in methylated cytosines was detected in seeds that were stored for one year at a 13.6% MC at −20 °C (R = 6.77%) and +3 °C (R = 6.17%), while the lowest level of 5mC (R = 5.49%) was observed in seeds stored at +10 °C (Figure 4c).

The highest level of 5mC in seeds from Seed Lot 2 at a 7.8% MC was observed in non-stored seeds (R = 10.1%). No significant changes in 5mC levels were observed in seeds with a 7.8% MC after storage for one year at −20 °C or −196 °C (Figure 4d). Storage at higher temperatures of +3 °C and +10 °C, however, resulted in a decrease in the level of 5mC (R = 8.22% and R = 7.09%, respectively) (Figure 4d).

A similar level of 5mC was observed in non-stored non-desiccated seeds from Seed Lot 3 at a 12.5% MC and seeds stored at −196 °C and −20 °C (ranging between 8.06 and 8.75%) (Figure 4e). Significantly lower levels of methylated cytosines were observed after one year of storage at +3 °C and +10 °C (R = 6.82% and R = 6.76%, respectively) (Figure 4e). For those seeds, a significantly decreased level of global DNA methylation was also observed after desiccation (Figure 4f).

### 3.5. Average Germination Rate of Populus nigra L. Seeds after One Year of Storage at Different Temperatures

No significant differences were observed in the average germination percentage between non-stored, control seeds and seeds stored for one year at −20 °C or −196 °C in any of the three seed lots (Figure 5a). Differences were observed, however, in the median germination values among the different seed groups. Storage of P. nigra seeds for one year at the higher temperatures of +3 °C and +10 °C resulted in a significant reduction in average germination rate down to 33.9% and 8.6%, respectively. The difference in the median values was also observed, with significant reductions to 8% and 0%, respectively (Figure 5a).

### 3.6. Average Level of DNA Methylation Level of Populus nigra L. Seeds after One Year of Storage at Different Temperatures

The highest average level of 5mC among all investigated seed groups was observed in non-stored, control seeds (R = 8.77% with a median value of R = 8.71%) and seeds stored in cryogenic conditions (R = 8.49% with a median value of R = 8.35%) (Figure 5b). Storage for one year at −20 °C resulted in a significant reduction in the global level of methylated cytosines to a value of R = 8.02%, with a median value of R = 7.99%. Storage of black poplar seeds at higher temperatures resulted in further significant decreases in the level of 5mC to R = 6.99% (with a median value of R = 6.89%) and R = 6.36% (with a median value of R = 6.31%) (Figure 5b).

### 3.7. Correlation between the Rate of Germination and Global 5mC Levels and Determination of Threshold 5mC Values Resulting in a Change in the Viability of Stored Seeds

For all seed lots, regardless of their MC, a significant correlation between the global level of 5mC and germination was detected as well as for non-desiccated and desiccated variants separately (Table 1). Statistical analysis revealed a positive and highly significant (*p* < 0.0001) correlation between the measured germination and global 5mC levels (R^2^ = 0.797), (Figure 6), as well for all data that were pulled together (Figure 5a,b).

Three highly significant models (Appendix A) were constructed based on the obtained data and p.dose function for prediction of the 5mC level at which the viability of seeds starts decreasing (Table 2). These three models were used to set a 5mC range of R = 7.638–8.208% as critical for maintaining the viability of *P. nigra* at approximately 80%. Then, based on the obtained data (Table 2), the relative changes (Δ%) in the 5mC level at different levels of germination were calculated (Appendix A; Figure 7). For all seeds, non-desiccated and desiccated, the change in 5mC percentage (Δ%) predicting viability decline ranged between 8–10%. Further reductions in the level of DNA methylation resulted in a rapid decrease in seed germination (Figure 7).

## 4. Discussion

Plant ex situ conservation methods are mainly based on seed storage. Aging processes in the intermediate seeds of exceptional EF3 plant species, however, preclude their long-term storage in seed banks as they suffer a steep decline in viability. Therefore, a better understanding of the process of seed aging is required to develop protocols that improve the conservation of seeds of such plant species [7,8,42]. It is essential to preserve seeds under conditions that do not cause a reduction in viability or foster genetic instability and epimutations. Plants have evolved mechanisms for repairing the damage that occurs to DNA in aging seed embryos during seed imbibition and germination, and this innate ability will determine seed vigor and viability [42,43]. The accumulation of DNA damages may also affect the pattern and degree of DNA methylation [44], a second level of information carried by DNA, which plays an important role in seed development [45], dormancy release [46,47,48], desiccation tolerance [36,37,38], and aging [30,31,33,34]. Therefore, maintaining and monitoring genetic and epigenetic stability is crucial for the successful storage of plant germplasm.

The overall objective of the present research was to identify the initial level of global DNA methylation change that precedes a decrease in seed viability. In other words, we aimed to test whether DNA methylation change can be used as an early indicator of the seed aging process in its asymptomatic phase, when no changes in germinability are observed. Therefore, we focused on measuring changes in global 5mC levels in *P. nigra* seeds in response to one year of storage at various temperatures and seed MC.

First of all, we observed that desiccation of black poplar seeds to an 8% MC induced a significant increase in the level of 5mC. Interestingly, a similar increase in methylated cytosines that did not alter the rate of seed germination was observed during the desiccation of orthodox seeds of *Pyrus communis* L., indicating that orthodox seeds can tolerate desiccation and can be successfully stored in a desiccated state under conventional conditions for a long period of time without adversely affecting their viability [38,49]. Previous research on the orthodox seeds of *Arabidopsis thaliana* revealed an increase in cytosine methylation in all DNA sequence contexts during the post-maturation to dry stages of seed development, indicating that RNA-directed DNA methylation (RdDM) is still active during desiccation until the seed enters dormancy [48]. Therefore, it can be assumed that RdDM is the main mechanism responsible for seed hypermethylation since cell division and DNA replication do not take place during the desiccation of seeds [48]. Increased DNA methylation during desiccation of *A. thaliana* seeds may foster chromatin packing to regulate gene expression and prevent transposable elements TE activation [48,50]. Importantly, global DNA hypomethylation may also release tightly packed chromatin and promote the expression of germination-related genes when conditions become favorable for germination [48,50]. This premise is supported by the fact that heterochromatic chromocentromers are smaller just after germination than they are at three weeks after germination and during germination [51,52]. Notably, the opposite trend in changes in the level of global DNA methylation has been observed in recalcitrant seeds of exceptional desiccation-sensitive species (EF2) *Acer pseudoplatanus* L. In those seeds, desiccation induced a decrease in the level of 5mC, which was significantly related to a decrease in their viability [37,53]. Therefore, our data show for the first time that the desiccation-induced epigenetic response of intermediate seeds produced by the exceptional plant species (EP 3) *P. nigra* is similar to the response observed in orthodox seeds.

Secondly, results revealed that the differences observed in the global level of methylated cytosines were related to seed quality. This was true for freshly harvested, non-stored seeds as seeds from Seed Lots 2 and 3 had higher germination capacity and higher 5mC level relative to freshly-harvested non-desiccated seeds from Seed Lot 1 (Figure 1 and Figure 2). As the aim was to verify if 5mC could be considered a universal molecular marker of seed viability, the correlation of germination and DNA methylation level of *P. nigra* seeds for all seed lots at high and low MC was tested. Indeed, in all tested seeds, lots of germination decline were significantly correlated with DNA methylation decrease regardless of the MC of seeds and their initial viability (Table 1). Such results indicate, that irrespective to anticipated mechanisms (enzymatic vs. driven by oxidative agents possibly released by the excited photosynthetic system) ongoing in hydrated or dried cytoplasm [13,16], a similar change in DNA methylation occurs during seed deterioration. This is in concordance with the latest research on recalcitrant *Acer pseudoplatanus* L. embryonic axes showing 5mC decline in tissues subjected to severe desiccation or accelerated aging [53]. Therefore, in the next step, all results for each seed lot were pulled together (Figure 5a,b). Consequently, *P. nigra* seeds exhibited a significant correlation (R^2^ = 0.797) between a decrease in their viability during storage and a decline in the level of 5mC (Figure 6). Notably, an identical correlation between a decreasing level of 5mC and germination (R^2^ = 0.7939) was observed in our previous research on recalcitrant seeds of *Quercus robur* L. [34].

To verify the hypothesis that 5mC measurement could serve as an early and universal indicator of seed aging initiation based on all of the results obtained on *P. nigra* seeds in storage as well as on separated data for non-desiccated and desiccated seeds, we constructed a relation model that predicts the decrease in seed germination capacity based on the percentage of changes in the level of global 5mC (Figure 7). Interestingly, the constructed relation models are similar to the model of seed aging in time [54] (Figure 6), where the phase of asymptotic change is followed by a rapid decline in viability. Consequently, we have demonstrated in the present study that once the progressive decline in cytosine demethylation crosses a determined 5mC threshold of approximately ~8–10% of change in relation to fresh seeds (non-desiccated, non-stored), a rapid decline in germination will occur. Importantly, after passing the threshold, the difference in the percent of 5mC change between non-desiccated and desiccated seeds is noticeable, as for non-desiccated seeds one percent of change in 5mC causes higher germination decline (Appendix A). This is consistent with the higher kinetics of the deterioration process at a higher MC [13,54,55,56]. Further research should focus on the identification of genome regions that are mostly affected by DNA demethylation causing germinability decline.

Consequently, even though a reduction in the level of 5mC was observed during storage of black poplar seeds at −20 °C (Figure 4), it did not exceed the ~8–10% level predicted to adversely impact germination capacity; therefore, seed viability remained unchanged (Figure 3). The decrease in the level of 5mC indicates, however, that the method of successful storage of orthodox seeds at −20 °C cannot be used for the long-term preservation of short-lived seeds of exceptional species (EF3), such as *P. nigra,* as it did not result in the maintenance of DNA methylation integrity (Figure 4 and Figure 5b).

Moreover, because a more complex experimental design was used in the present study as different storage regimes (seed MC × storage temperature), our present results support the validity of our previous research [49]. Specifically, these data demonstrate that storage under cryogenic conditions at an optimal MC (Figure 4) helps to maintain the epigenetic stability of seeds and also shows that storage at temperatures above 0 °C (+3 °C and +10 °C) induces a decrease in the average level of global DNA methylation by 20.3% and 27.5% relative to the non-stored seeds (Figure 5b). These data indicate that to maintain epigenetic integrity, it is essential to store seeds at an appropriate MC and temperature. It can be also assumed that the demethylation observed at higher temperatures is due to active demethylation via the base excision repair (BER) process catalyzed by DNA demethylases when 5mC is directly removed [50] or thymine-DNA glycosylase (TDG) when the potential oxidation products of 5mC (5-hydroxymethylcytosine, 5-formylcytosine, and 5-carboxycytosine) are recognized and cleaved [53]. Regarding the latter, an increase in ROS (O_2_^−●^ and H_2_O_2_) was observed in *P. nigra* seeds after one year of storage at +3 °C and −20 °C [27]. To reiterate, the ~8–10% 5mC threshold was valid regardless of the initial global 5mC level and germination capacity, as a decline of ~8–10%, was still needed to reduce germination. Exceeding the ~8–10% 5mC threshold may impair epigenetic stability, leading to a reduction in germination capacity. A 40% decline in DNA methylation relative to the control appeared to be lethal as residual germination was observed in these seeds (Figure 7). It seems that the integrity of genomic DNA cannot be maintained at this stage and that efficient and accurate DNA repair is no longer possible [57]. Indeed, seeds have developed a repair system that allows them to minimize damage and repair biological molecules and cellular structures during imbibition; therefore, the ability of seeds to repair themselves is closely linked to their germination capacity [58].

## 5. Conclusions

Aging is a complicated biological phenomenon which is associated with diverse biochemical, molecular metabolic, and physiological processes [58]. Our present study demonstrated the importance of the global DNA methylation status in the seed response to desiccation and aging. Although previous studies have also reported that changes in the 5mC profile are associated with a decline in seed viability in orthodox and recalcitrant seeds [30,31,34,59,60], the usage of short-lived intermediate seeds produced by exceptional EF3 species gave us the opportunity to investigate the seed deterioration process without application of accelerated aging protocols, thereby corresponding to natural aging in dry and cold storage. This seems to be important as short seed lifespan is considered to be a common trait in a majority of species in diverse collections and storage conditions [13,61]. The current research was focused on DNA methylation analysis; however, it is important to keep in mind that other processes, such as post-translational histone modifications and regulation imposed by ncRNA, are also involved in epigenetic regulation of genome structure and gene expression in seeds [42].

Early studies have already shown that chromosomal damage, such as breaks in DNA, DNA methylation, and abnormal gene expression, may accumulate during the process of seed aging, causing the frequency of affected cell damage to surpass a threshold after which seeds lose their germinability [58]. Moreover, although epigenetic modifications can be reset between generations, the transmission of gradually acquired epimutations over generations appears to be higher in plants than in animals [62,63,64,65]. Epigenetic alterations, especially those involving DNA methylation, may contribute to transgenerational epigenetic variation. In fact, DNA methylation has been reported to be involved in the epigenetic regulation of phenotypic variation in *P. nigra* seedlings [66]. Therefore, it is crucial to preserve the epigenetic landscape in stored seeds as well as preserve their viability to ensure that true-to-type plants can be generated when seeds are germinated. We demonstrated that when a critical threshold of an ~8–10% decline in 5mC in comparison to control fresh seeds is achieved during storage (aging), seed viability begins to decline regardless of both initial seed viability and their MC. Even though the mechanisms behind DNA demethylation at hydrated and dry cytoplasm still need to be investigated, we were able to demonstrate that changes in the global level of 5mC analyzed in DNA isolated from entire poplar seeds precede a decline in seed viability. Therefore, tracking the level of 5mC in the genomic DNA of seeds would enable one to identify the moment that rapid aging is initiated, allowing one to regenerate seeds before the change in the level of 5mC reaches ~8–10%. It appears that such an approach can be used as a molecular marker of seed viability and that its use would greatly benefit the preservation of seeds of exceptional species.

We also demonstrated that after *P. nigra* seeds are shed, they still exhibit an increase in DNA methylation in response to desiccation. Comparable to orthodox seeds, this may be considered an adaptive mechanism to environmental stress and water withdrawal [36,37,67].

## Figures and Tables

**Figure 1 cells-11-02080-f001:**
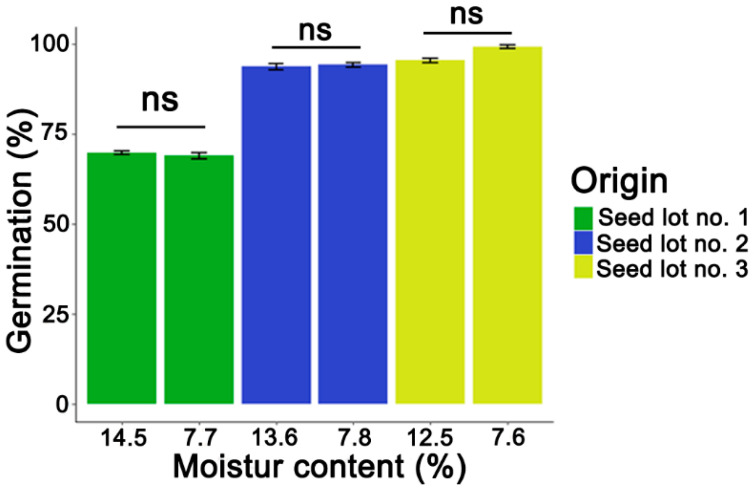
Germination of non-stored non-desiccated and desiccated *Populus nigra* L. seed lots. Statistical analysis was conducted using the Wilcoxon–Mann–Whitney test. Values marked with the “ns” are not significantly different at *p* < 0.05. Data represent mean ± SE, *n* = 4.

**Figure 2 cells-11-02080-f002:**
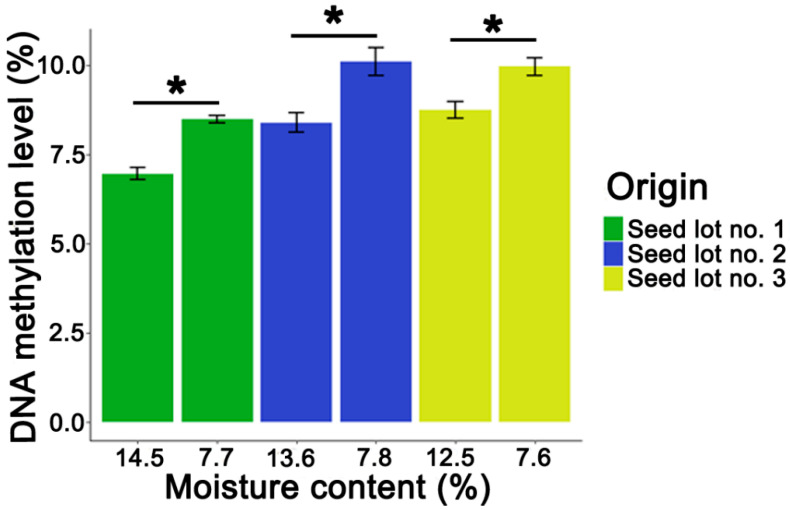
Global 5-methylcytosine level in non-stored non-desiccated and desiccated *Populus nigra* L. seed lots. Statistical analysis was conducted using the Wilcoxon–Mann–Whitney test. Separate statistical analysis was conducted for every seed lot. Values marked with * are significantly different at *p* < 0.05. Data represent mean ± SE, *n* = 4.

**Figure 3 cells-11-02080-f003:**
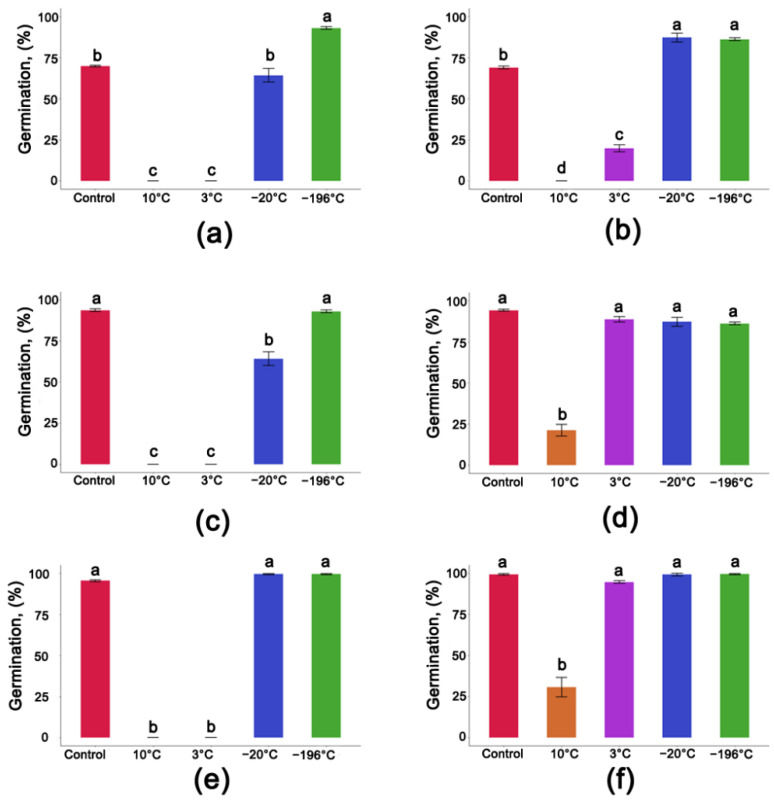
Germination of *Populus nigra* L. seeds stored for one year at different MCs: (**a**) Seed Lot 1 stored at MC of 14.5%; (**b**) Seed Lot 1 stored at MC of 7.6%; (**c**) Seed Lot 2 stored at MC of 13.6%; (**d**) Seed Lot 2 stored at MC of 7.8%; (**e**) Seed Lot 3 stored at MC of 12.5%; (**f**) Seed Lot 3 stored at MC of 7.6%. Controls were composed of non-stored seeds. Statistical analysis was conducted on a GLM model with a binomial distribution using ANOVA and a Duncan test. Values marked with the same letter are not significantly different at *p* < 0.05. Data represent mean ± SE, *n* = 4.

**Figure 4 cells-11-02080-f004:**
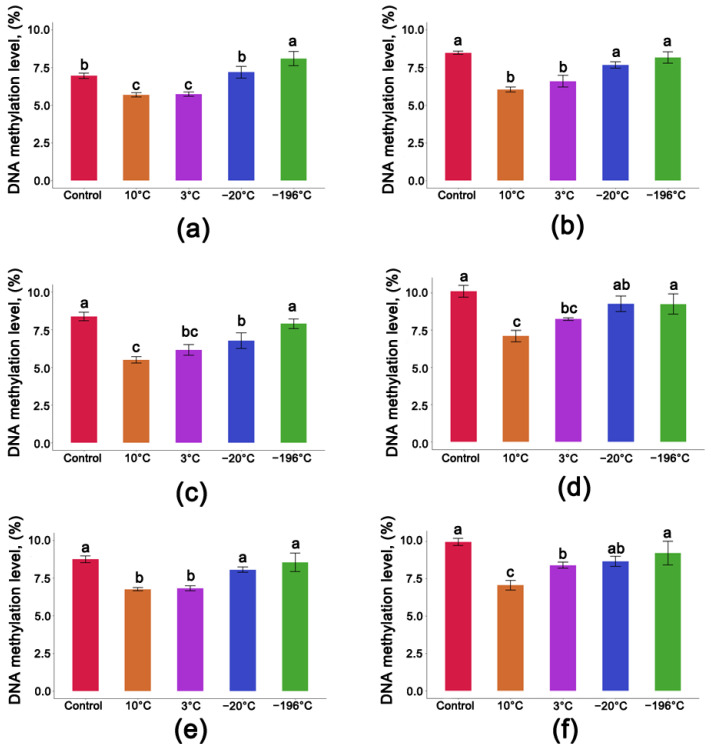
DNA methylation level of *P. nigra* seeds stored for one year at different MCs: (**a**) Seed Lot 1 stored at MC of 14.5%; (**b**) Seed Lot 1 stored at MC of 7.6%; (**c**) Seed Lot 2 stored at MC of 13.6%; (**d**) Seed Lot 2 stored at MC of 7.8%; (**e**) Seed Lot 3 stored at MC of 12.5%; (**f**) Seed Lot 3 stored at MC of 7.6%. Controls were composed of non-stored seeds. Statistical analysis was conducted using ANOVA and a Duncan test. Values marked with the same letter are not significantly different at *p* < 0.05. Data represent mean ± SE, *n* = 4.

**Figure 5 cells-11-02080-f005:**
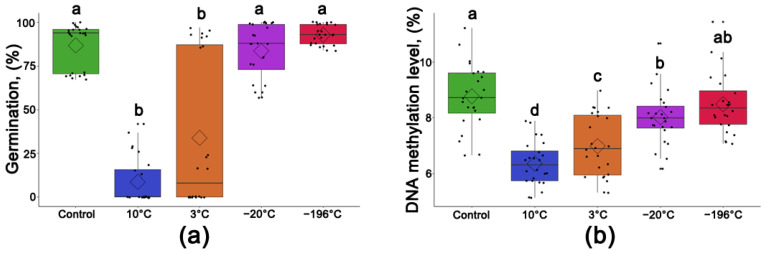
Average germination capacity (**a**) and levels of 5mC (**b**) of non-stored and stored, desiccated and non-desiccated *Populus nigra* L. seeds pulled from three seed lots. Control was composed of non-stored seeds. Separate statistical analyses were conducted for germination and 5mC level data. Statistical analysis was conducted on a GLM model with a binomial distribution (**a**) or LM (**b**) followed by ANOVA and a Duncan test. Values marked with the same letter are not significantly different at *p* < 0.05. Data represent mean ± SE, *n* = 24.

**Figure 6 cells-11-02080-f006:**
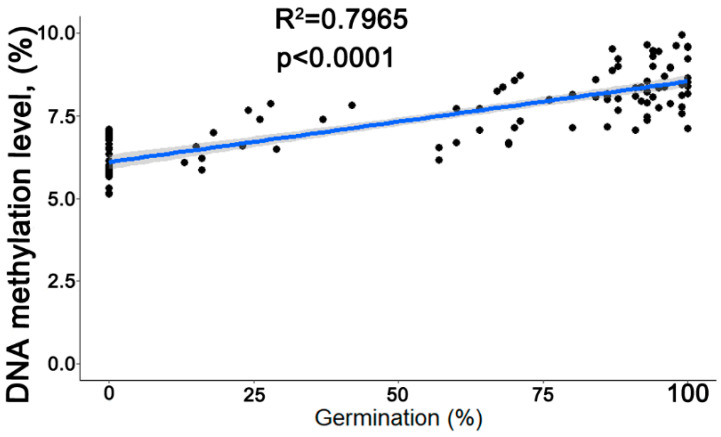
Correlation between germination and global 5mC level for three combined seed lots composed of non-desiccated and desiccated seeds. The grey area represents the level of confidence interval at 0.95 (*n* = 120).

**Figure 7 cells-11-02080-f007:**
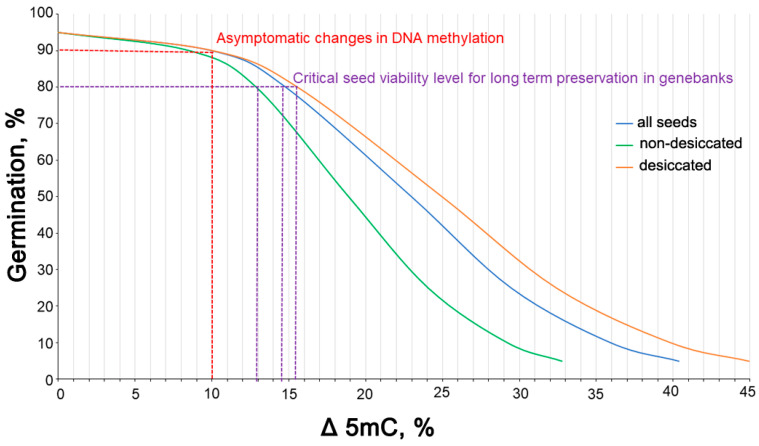
The relationship between germination capacity (%) of *Populus nigra* L. seeds and relative percentage of decline (Δ%) at 5mC level for: all seeds (dark blue, n = 120, AIC = 73.605); non-desiccated seeds at the MC range of 12.5–14.5% (green, N = 60, AIC = 41.184); and desiccated seeds at the MC range of 7.6–7.8% (orange, N = 60, AIC = 34.459). As a reference data, the percentage of 5mC at a germination level of 95% calculated with the dose.p function in R was used. All GLM model coefficient values are provided in Appendix A.

**Table 1 cells-11-02080-t001:** Correlation of germination (%) and DNA methylation level (%) of *P. nigra* seeds.

Seed Lot	Moisture Content (%)	Correlation Coefficients	Correlation Significance
Seed lot No. 1 ^#^	14.5	0.87	*p* < 0.0001
Seed lot No. 1 ^#^	7.6	0.822	*p* < 0.0001
Seed lot No. 2 ^#^	13.6	0.84	*p* < 0.0001
Seed lot No. 2 ^#^	7.8	0.723	*p* = 0.0003
Seed lot No. 3 ^#^	12.5	0.802	*p* < 0.0001
Seed lot No. 3 ^#^	7.6	0.694	*p* = 0.0007
Non-desiccated *	12.5–14.5	0.822	*p* < 0.0001
Desiccated *	7.6–7.8	0.775	*p* < 0.0001

^#^*n* = 20, * *n* = 60.

**Table 2 cells-11-02080-t002:** Critical levels of germination and corresponding 5mC levels calculated according to a p.dose function constructed on GLM models on combined data from all seeds, and separate data from non-desiccated seeds at MC 12.5–14.5% and desiccated seeds at MC 7.6–7.8%.

Assumed Germination Level	Predicted DNA Methylation Level (%) ^1^		
	All Seeds	Non-Desiccated Seeds	Desiccated Seeds
95%	9.375 ± 0.382	8.767 ± 0.435	9.859 ± 0.48
90%	8.347 ± 0.237	8.018 ± 0.28	8.742 ± 0.369
80% *	7.989 ± 0.181 *	7.638 ± 0.216 *	8.208 ± 0.274 *
50%	7.221 ± 0.147	7.111 ± 0.161	7.294 ± 0.265
25%	6.613 ± 0.198	6.657 ± 0.197	6.57 ± 0.394
10%	6.005 ± 0.283	6.204 ± 0.278	5.845 ± 0.565
5%	5.591 ± 0.347	5.895 ± 0.344	5.353 ± 0.69

* Viability critical for storage at gene banks. ^1^ Data significant at the level of *p* < 0.0001.

## Data Availability

The data presented in this study are available on request from the corresponding author.

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
