# Peer review of "DNA Methylation as an Early Indicator of Aging in Stored Seeds of “Exceptional” Species Populus nigra L."

_cells, 2022, doi:10.3390/cells11132080_

Round 1
Reviewer 1 Report
The authors investigated the DNA methylation profile of P. nigra after dessication and storage by evaluating the correlation with germination. I advise that minor corrections should be made in the paper to be considered for publication in Cells. The manuscript pdf contains the corrections that should be addressed by the authors.
My major concerns are:
a) The correlation among the DNA methylation changes and maintenance of epigenetic integrity. As we have other epigenetic mechanisms involved in epigenetic regulation, the authors should not attribute only to DNA methylation the role of epigenetic integrity, since at least two mechanisms are part of this process - small RNAs and histone modifications.
b) To evaluate the percentage of seeds germination, the authors considered the emerging radical, collet hairs, and fully expanded cotyledons after 14 days. However, they did not provide any biometric data about these seedlings. If possible, I would recommend the addition of this data, once the storage could impact not only the germination step but also the seedlings growth.

Author Response
Response to Reviewer 1
We have addressed all suggestions made by Reviewer. We would like to thank the Reviewer for recognition of our work and contributions to perfecting the manuscript. We hope to get the Reviewer’s approval.
Comment: The abbreviation should be 5mC
Our response: The suggestion was acknowledged and the corrections were made.
Comment: Populus nigra L. (black poplar) is one of the most endangered tree species in [13].
Our response: Corrected. Current version: Populus nigra L. (black poplar) is one of the most endangered tree species in Europe [13].
Comment: It is important to cite here works that evaluate DNA methylation in seeds during desiccation and storage (e.g. Michalak et al, 2013; Mira et al, 2020).
Our response: This suggestion was acknowledged and entire paragraph was added with required citations and description of previously published research. Please see manuscript with tracked changes.
Comment: Please add the information that control is composed of non-stored seeds.
Our response: Such information has been added into figure captions.
Comment: Subjective term. Please remove.
Our response: Removed.
Comment: Please add the reference here.
Our response: Corrected.
Comment: The correlation among the DNA methylation changes and maintenance of epigenetic integrity. As we have other epigenetic mechanisms involved in epigenetic regulation, the authors should not attribute only to DNA methylation the role of epigenetic integrity, since at least two mechanisms are part of this process - small RNAs and histone modifications.
Our response: Authors agree with this statement, therefore additional fragment has been introduced into Conclusions section:
“The current research was focused on DNA methylation analysis, however, it is important to keep in mind that also other processes, as post-translational histone modifications and regulation imposed by ncRNA, are involved in epigenetic regulation of genome structure and gene expression is seeds [42].”
Comment: To evaluate the percentage of seeds germination, the authors considered the emerging radical, collet hairs, and fully expanded cotyledons after 14 days. However, they did not provide any biometric data about these seedlings. If possible, I would recommend the addition of this data, once the storage could impact not only the germination step but also the seedlings growth.
Our response: Unfortunately, only total germination of seeds was evaluated in our experiments, therefore we can not provide any additional data about seedling growth.
Reviewer 2 Report
This is an interesting manuscript on the role of DNA methylation on seed ageing during dry storage at low temperatures and on how this measure could be used as an early indicator of seed aging. The authors used for their research P. nigra, a species producing seeds with extraordinary short lifespan even at conventional seed bank conditions, aiming also to contribute to improve the storage and monitoring options for seeds of this exceptional species. In my opinion, the short lifespan of the seeds used is a great choice and an advantage for this research, as it allowed the authors to investigate the role of DNA methylation on seed ageing at storage conditions relevant to seed banking (dry and cold) within a year. On the contrary, previous research has relied in accelerated ageing approaches (wet and warm), which may not be ideal to determine the exact mechanisms of ageing during dry and cold storage. While the research idea is great and most of the methods used seem soundly, I found some issues that need to be attended by the authors before publication. Firstly, the introduction lacks a proper presentation of the state of art of the research topic: DNA methylation and seed ageing. This lack of background and previous findings around the role of DNA methylation on seed ageing is imperative for two reasons: (1) it is needed for the reader to understand the background and the novelty of the research work presented, and (2) it will help to the authors to build the research hypotheses to test in the research. Here the second flaw, the paper does not present any hypothesis to test even though the authors have investigated the same topic before in almost a half-dozen species. Hypotheses to test are needed in a research paper. The third issue I found is the repetition of some of the data presented in the results section, and the lack of strategy (based on hypotheses) to determine when the data from all seeds collected and all treatment assayed is pulled together, when is presented for individual trees (seed lots), or when is presented for all combinations of seed lots and treatments. Also, the stats used to analyse data that follows a binomial distribution must be revised (particularly ANOVA and Duncan). And finally, in my opinion, a big flaw is the predictive model constructed to determine the methylation levels of a seed at specific germination percentages. For example, the development of this model (and the data used to build it) is not well explained in the methods. One can intuitively understand what is the origin of the data used for the correlation shown in Fig 7. But the way figure 8 has been constructed is completely unclear (particularly when the control methylation values varied across seed lots and drying conditions). In addition, the model do not show the variation or the confidence interval, and the predictive power of the model is unclear, as a probability of prediction based on the variability of the data is not provided. This lack of explanation on how the model has been built, how a sigmoidal trend is obtained in Fig 8 (if fig 7 shows a linear trend of the raw data), and the low power of the model may interfere in the conclusions reached by the authors in terms of the use of the methylation level as an early indicator of seed aging. The interpretation of the data in this respect is crucial for this paper based on the title presented.
With independency of the flaws indicated I think the data set presented is novel, very interesting and has a great potential, and I encourage the authors to improve the manuscript around the issues indicated above: (1) a good background on DNA methylation in seed ageing, (2) a set of hypotheses to test, (3) a proper and non-repetitive way to present data taking into consideration all variables (seed lot, drying level and temperature), and (4) a solid and well explained predictive/descriptive method to confirm why DNA methylation levels are an early indicator of seed aging. Other minor comments on the manuscript are also included.
Title: The stored material in which DNA methylation is an early indicator of ageing are the seeds, not the “exceptional” plant species used for the experiments (Populus nigra). I think it would be more accurate if the title is edited a bit, and reads as: “DNA methylation as an early indicator of aging during the storage of the seeds of the “exceptional” species Populus nigra L.” or “DNA methylation as an early indicator of aging in stored seeds of the “exceptional” species Populus nigra L.”
Abstract: Authors are using the scientific name of the species through the whole abstract, except in line 28 “Once poplar seeds reached a 10% reduction in the level of m5C”. For consistency and to avoid confusion to the reader (who may not be familiar with the common name of the species and this has not been defined in the abstract) I would use the term “P. nigra” instead of “poplar” in this particular sentence of the abstract: “Once P. nigra seeds reached a 10% reduction in the level of m5C”.
Introduction:
Line 38, where it says “conversation” should say “conservation”.
Lines 49-51 “seed banking is based on desiccating seeds at 15% of relative humidity (RH) to a specific moisture content (MC) followed by storage at -20 °C”. In this sentence, if authors are talking about the RH for desiccation before storage, I recommend following the FAO standards, that indicate that drying can be done at a range of RHs: 10-25 % (and a range of temperatures: 5–20 °C). The 15% RH indicated for drying in seed banking, although correct, is not the standard in all seed banks (see, e.g., seed banks in the CPC network), and the sentence should be more inclusive of other drying conditions supported by FAO standards. However, if authors want to indicate that the RH at the storage temperature should be 15%, then this would ok as per FAO standards, but the sentence should be edited accordingly.
Paragraph between lines 60-64. I fully agree with this paragraph, and I am glad the authors mention the issue of diversity of ageing mechanism and the use of just a few model species to study seed ageing. However, I would extend the discussion on this issue a bit more (maybe one more sentence here or in the discussion section). For example, ageing may be dependent on the physical-chemical characteristics of the cells in the seeds (so called dry architecture in Ballesteros D, Pritchard HW, Walters C (2020). Dry architecture: towards the understanding of the variation of longevity in desiccation-tolerant germplasm. Seed Science Research 30, 142–155. https://doi.org/10.1017/S0960258520000239). This is particularly relevant for this paper, as for example Arabidopsis seeds (the model species mentioned by the authors here) are oily seeds in which ageing may be related to the oxidation or/and crystallization/melting issues induced by/ produced in the storage lipids, and Populus nigra seeds contain developed chloroplasts in the mature seeds that may modulate their ageing during storage (see conclusions in Roqueiro G et al. 2010. Effects of photooxidation on membrane integrity in Salix nigra seeds. Annals of Botany 105: 1027–1034, doi:10.1093/aob/mcq067, and comments on ageing of chlorophyllous seeds in Ballesteros et al., 2020). Could the authors somehow introduce this diversity of ageing mechanisms depending on the physical-chemical properties of their cells (or dry architecture) here? This is also relevant for the next paragraph when authors talk about ageing mechanism studied in P. nigra seeds. There, the potential damaging effect of the chloroplasts in the dry cells of the seeds has not been introduced. This is a suggestion for the edition of the text (edition in red) in case the authors want to take my comment/suggestion into consideration: “Fundamental knowledge about the progressive loss of seed viability during storage is based on studies conducted in crops and model species (Arabidopsis thaliana L.) which produce orthodox seeds of relatively long lifespan [9–12]. In the case of short-lived seeds, however, our knowledge of the factors determining seed viability is insufficient, which prevents the establishment of effective methods for long-term storage. In this respect, it has recently been suggested that ageing mechanisms may be different in seeds of different species based on their cell’s “dry architecture”, particularly in relation to seeds with short lifespan [Ballesteros et al., 2020]. For example, P. nigra seeds have very low storage lipids and contain developed chloroplasts in their dry mature cells, which may be the main source of oxidative stress inducing the fast ageing in these seeds [Roqueiro et al., 2010; Ballesteros et al., 2020]. On the other hand, A. thaliana seeds (one of the model species mentioned above), do not contain chloroplasts and have relatively high amounts of storage lipids that, instead of the chloroplast, may be more involved in the slower ageing of the seeds in this and other oily species [Ballesteros et al., 2020; Zinsmeister et al., 2020; Gerna et al., 2022]”. Zinsmeister, J., Leprince, O., & Buitink, J. (2020). Molecular and environmental factors regulating seed longevity. Biochemical Journal, 477(2), 305-323. Gerna, D., Ballesteros, D., Arc, E., Stöggl, W., Seal, C. E., Marami-Zonouz, N., ... & Roach, T. (2022). Does oxygen affect ageing mechanisms of Pinus densiflora seeds? A matter of cytoplasmic physical state. Journal of Experimental Botany, 73(8), 2631-2649.
Line 65: “Populus nigra L. (black poplar) is one of the most endangered tree species in WHERE? [13]” The place where P. nigra is one of the most endangered tree species is not indicated. It is the World? Europe? France? Poland?
Lines 76-80: If not detailed above as suggested, I would add to this enumeration of physiological and biochemical measures of ageing in P. nigra seeds, the potential negative role of the chloroplasts that are part of the dry mature seeds stored. For example (editions or additions in red): “The physiological and biochemical changes of P. nigra seeds in relation to ageing have been the subject of several studies [13,18–23]. Specifically, these studies have demonstrated that storage of black poplar seeds results in increased oxidative stress [22], large changes in protein abundance [23], and a reduced proportion of roots with absorptive function [21]. Interestingly, many of these changes could be induced by an exacerbated oxidative stress mediated by the chloroplasts found in the dry mature seeds of this species [Roqueiro et al., 2010; Ballesteros et al., 2020]”
Paragraph between lines 83-87. DNA methylation in seeds during ageing is the main topic of this paper, but the background on this topic has not been sufficiently introduced. A reader not familiar with the topic will need this information to understand the novelty of the research around the topic. Also, authors must introduce the state of the art in the topic “DNA methylation in seeds during ageing” so proper hypothesis for the research can be established. This is particularly relevant when the authors have investigated this topic previously in diverse species: see references cited in this manuscript [26, 27, 28, 36, 38]. In addition to their own research, I recommend adding information from the recent research made by other authors in this topic, like Mira et al (Mira, S., Pirredda, M., Martín-Sánchez, M., Marchessi, J. E., & Martín, C. (2020). DNA methylation and integrity in aged seeds and regenerated plants. Seed Science Research, 30(2), 92-100), Pirreda et al (Pirredda, M., González-Benito, M. E., Martín, C., & Mira, S. (2020). Genetic and epigenetic stability in rye seeds under different storage conditions: Ageing and oxygen effect. Plants, 9(3), 393), and Yalamalle et al (Yalamalle, V. R., Ithape, D. M., Kumar, A., Bhagat, K., Ghosh, S., & Singh, M. (2020). Seed treatment with 5-azacytidine reduces ageing-induced damage in onion seeds. Seed Science and Technology, 48(3), 407-412.).
Lines 88-90: Authors state that “The objective of the present research was to determine if changes in global m5C levels affect or are correlated with the viability of short-lived, intermediate seeds produced by exceptional species.” However, based in previous findings of the authors and the other research on seed DNA methylation presented above, this objective is a bit trivial. Authors must set (at least) one novel hypothesis to test, particularly when they have investigated this topic before in diverse orthodox, intermediate, and recalcitrant seeded species [26, 27, 28, 36, 38] and there is recent research published in diverse orthodox seeded species [Mira et al., 2020; Pirredda et al., 2020, Yalamalle et al., 2020]. What is new in this research? For example, authors use dry vs moist storage at different low temperatures while Mira et al 2020 and Pirredda et al 2020 did their research on high temperature and moisture conditions. Or which previous unresolved questions are authors using to build up this research? For example, changes in DNA methylation during ageing are known, but their value as an early indicator of seed aging has not been proved. If hypotheses are not presented, the paper seems just the repetition of a method but in a different species. Authors must introduce an advance to this topic previously investigated in half-dozen species. Maybe a way to start is to transform the objective of the first sentence and the overall goal in the last sentence into proper hypotheses to test: “We hypothesize that changes in global m5C levels will be correlated with the viability of short-lived, intermediate seeds produced by the exceptional species P. nigra during dry and cold storage. In addition, we hypothesize that changes in DNA methylation (m5C) will be observed prior to the main reduction in seed viability”. But based on the author’s experience in this topic, I am sure they will have plenty of hypotheses they aimed to test with this research.
Methods:
Lines 96-98: Seeds were collected from three different trees in the same location near Czeszewo (52°8’N, 17°30’E), although separated 3 km one to each other. Was each seed collection (representing one single tree) considered as a “seed lot” for the data analyses? Please specify in the methods if this was the case. However, to me, these three trees seem like the remnants of a previously larger and now fragmented population. Is this Ok? Or are they three trees in a large population formed by hundreds of trees? Or three trees from three different populations? Or are they the single remnants of three different “near extinct” populations with just one or a few trees? Please clarify in the methods. This is important to consider the way to analyse the data. For example, if the three trees belong to a single population, why the seeds from the three trees were not pulled together and the three-tree collection was considered as a single collection? On the other hand, if authors wanted to study inter-tree differences within a population (maybe small and highly fragmented), this should be stated in the introduction, when setting the objectives/hypotheses, otherwise the reasons to treat the seed collection of each tree separately makes no sense to me.
Line 108: What is a “Jacobsen apparatus”? Please provide trademark and model and if possible, reference stating its use for seed germination previously, as not all readers may be familiar with this apparatus.
Line 145: In the sentence “The effect of desiccation and storage on seed germination”, does “storage” relate to “temperature”? If so it should read as: “The effect of desiccation and storage temperature on seed germination”. Same in line 147 “impact of desiccation and storage temperature on the level of DNA methylation”.
Lines 145-146: Here it is indicated that germination data for the GLM was treated as data following a binomial distribution. I suppose that this is due to the fact that each seed placed to germinate was considered as a “sample” and data was treated in a binomial way as success (germination) and no success (no germination) to build a proportion. Is that ok? But authors mention in lines 108-110 that 50 seeds were germinated in 4 technical replicates. Does this mean that the data from the 200 seeds (in 4 technical replicates) used for each conditions tested were pulled together for the data analysis in the GLM? Please specify the treatment of the data and replicates.
Lines 151-155: In line 145-146 is indicated that germination data was considered to follow a binomial distribution. However, lines 151-155 indicate that different tests were used to determine differences in the mean of germination across different drying and storage conditions. The tests used as per figures 1, 3 and 5 ANOVA and Duncan tests. However, these tests are suitable for data that follow a normal distribution, and not for the germination data that do not follow a normal distribution (indeed they were considered to follow abinomial distribution). Here there is a methodological problem. If authors want to determine differences in the mean of germination across different drying and storage conditions using ANOVA and Duncan test the raw germination % data must be transformed to approach the data into a normal distribution (e.g., by the arcsine transformation). Otherwise, proper “binomial tests” to account for differences in the proportion of germination across different drying and storage conditions must be used (e.g., Z test, Chi-square, Fischer,… here some indications for the Z-test: http://www.sthda.com/english/wiki/two-proportions-z-test-in-r )
Lines 161-164: The way this prediction was done needs to be better explained. For example, what was the original data used to construct/model the relation between germination and m5C level (table 2)? Was the data shown in the correlation expressed in Fig. 7? This means that data from all three seed lots and drying conditions was pulled together, isn’t it? Are not the authors curious about how the desiccation levels affect the correlation of methylation levels vs germination%? I am, though, as ageing mechanisms may be different in the non-dried and the dried seeds. BTW, when modeling this relation, what was the link used in the GLM? Logistic? Probit? Linear? It looks linear from data in Fig. 7, but this needs to be well defined in the methods. Finally, what data was used for constructing Figure 8? It is not clear in any place. For example, what was the reference (control) value of methylation used to calculate the relative percentage of change in m5C level in relation to control? If this value changed depending on the seed lot used and the desiccation level (Fig. 4), how was this done? Why figure 8 lacks variation in the data set? Supplementary information with all these calculations would be helpful to evaluate the model constructed.
Results:
Line 167: P. nigra needs to be in italics
Line 168: Each seed lot are seeds from a single tree, then I am not sure if the term “seed lot” is correct or it should just say “tree”.
Lines 169-170, In the sentence “Control seeds did, however, exhibit a difference in initial seed germination capacity (Figure 1)”, are the authors stating that the seeds from each tree (seed lot) showed a different germination % before drying? Maybe the sentence can be edited for clarity.
Line 176: Authors use diverse terms to refer to the same treatment: freshly collected seeds, control seeds, non-desiccated seeds, seed lot at a 14.5% of MC,… to avoid confusion and facilitate the reading and interpretation of results, please, be consistent with the way this treatment is referred along the text (results and discussion). The use of “control seeds” is not recommended as it is used with different meanings in different sections of the manuscript: e.g., control seeds are referred to non-desiccated seeds (line 169), but also to non-stored seeds (although they may have been desiccated: line 208).
Lines 187-213 (section 3.3): I think this section can be summarized and redundancies can be removed (particularly when data is shown in Fig 3), focusing the description of the results in the common patterns observed. The seeds from the three seed lots (trees) responded to the combination of moisture and temperature in the same way, although with some differences in % germination likely due to the different initial germination of each lot/tree. In all three seed lots, the absence of desiccation resulted in very low or nil germination at +10 and +3C, and often (2 out of 3 lots) a small reduction of germination after storage at -20C. On the other hand, desiccation resulted in high germination (similar to non-stored seeds) after storage at -20C and +3C (except for seed lot 1). Germination after storage at -10 was also low, but higher than that of non-desiccated seeds. Germination after LN did not show any significant decrease independently of the drying treatment. Interestingly germination after LN storage increased a bit in lot 1.
Lines 221-252 (section 3.4): Similarly to the comment made above, I think this section can be summarized, focusing the description of the results in the common patterns observed (which are in relation to the common patterns observed in section 3.3).
I am confused about sections 3.5 and 3.6, corresponding to figures 5 and 6. Why now data from figures 3 and 4 is presented again but without considering the desiccation level or seed provenance (lot/tree)? What is the value of presenting data in a different way if the message is the same but the desiccation component is lost? If the seed provenance changes the results observed, I would understand that data for each desiccation level and temperature combination is presented for “all seed collection (data from the three trees pulled)” vs “each tree harvested (the three seed lots used in the paper)”. However, this is not the case. Hence, I think sections 3.5 and 3.6 are redundant and should be removed.
Line 295 and 297: P. nigra needs to be in italics
Table 1 and Figure 7. Again, I do not understand why data is presented for all seed collection vs each single tree. Figure 7 is sufficiently self-explicatory for the correlation found. If the inter-tree differences were an objective of the paper and there were hypotheses to test in this regard, I understand data is presented in these two ways. But this is not the case based on the introduction of the paper.
Table 2: why germination % does not include a measure of variance as indicated for the methylation level?
Lines 296-301 and Figure 8: I am not convinced about the model presented here and the interpretations made on the value of the methylation levels to be used as an early indicator of seed aging. First of all, the methodological procedure to define the model is not well explained in the methods (already indicated above). In addition, the “relative decline in methylation levels” (X- axis in Fig. 8) is very confusing: where this relative value comes from? Was it individually calculated for the relative change in methylation at each storage temperature respect to non-stored seeds (control)? Then, did authors considered each combination of seed lot and moisture level (for which control values were different)? If so, why data points in figure 8 do not show any value of variation? For example, at relative decline in methylation “zero” (so the control values), germination % ranges from 75% (seed lot 1) to 93% (seed lot 2) and figure 8 shows a single point at 95%. A supplementary table showing all these calculations would help, and also the variation of the raw data source around the fitted sigmoidal line. To me, the value of figure 8 seem redundant (or artificial) when we have the raw correlation of the germination vs methylation and the estimated doses (methylation levels) for different germination % (which is extracted from the fitted correlation of Fig. 7 but using the dose.p function in R). With this correlation (figure 7) authors could estimate the probability of obtaining certain ranges of germination % based on the raw methylation levels. But I understand this raw correlation is not showing the early detection of seed ageing as methylation change, because the correlation between germination vs methylation (raw values in Fig 7) is linear (not sigmoidal). With a linear correlation is difficult to ascertain a level of methylation for which germination % changes but the methylation levels do not change (this would have been clear if the correlation is sigmoidal or composed by two linear trends). Fig. 8 is sigmoidal, I agree, but as said above, the source of the data used to make this figure is not clear. This figure is not built based on raw data or we would see larger dispersion of data points as in Fig 7. The construction of this model must be clarified and values of variation or probability need to be included.
Discussion:
A discussion of a research paper use to be written to discuss if the results showed in the results section support the hypotheses presented in the introduction. But since no hypotheses are presented in the introduction of this paper, the points to be discussed seem arbitrary. Nonetheless, the discussion raises some important issues that could conform some of the hypotheses that are lacking in the paper. For example, the importance of interspecific seed quality (from different populations or single trees/lots, as is the case here) on the initial methylation levels (lines 331-333) but how the relative changes in methylation during seed ageing are independent of these initial levels (lines 395-397). Or if there is an interaction in the levels of methylation (or their relative decrease) during storage at different temperatures based on the hydration levels (said in other words, does methylation change in the seeds in a different way when they are dry or wet?). In my opinion, the discussion will have to be rewritten based on the hypotheses presented.
Lines 385-402. This is a fundamental piece of the paper. However, to validate this interpretation, the model constructed (Fig. 8) need to be clarified and the measures of variation need to be included.
Author Response
Response to Reviewer 2
We would like to thank the Reviewer for the contribution to perfecting the manuscript. We have improved the manuscript according to the suggestions. We hope to get the Reviewer’s approval.
Comment: This is an interesting manuscript on the role of DNA methylation on seed ageing during dry storage at low temperatures and on how this measure could be used as an early indicator of seed aging. The authors used for their research P. nigra, a species producing seeds with extraordinary short lifespan even at conventional seed bank conditions, aiming also to contribute to improve the storage and monitoring options for seeds of this exceptional species. In my opinion, the short lifespan of the seeds used is a great choice and an advantage for this research, as it allowed the authors to investigate the role of DNA methylation on seed ageing at storage conditions relevant to seed banking (dry and cold) within a year. On the contrary, previous research has relied in accelerated ageing approaches (wet and warm), which may not be ideal to determine the exact mechanisms of ageing during dry and cold storage. While the research idea is great and most of the methods used seem soundly, I found some issues that need to be attended by the authors before publication. Firstly, the introduction lacks a proper presentation of the state of art of the research topic: DNA methylation and seed ageing. This lack of background and previous findings around the role of DNA methylation on seed ageing is imperative for two reasons: (1) it is needed for the reader to understand the background and the novelty of the research work presented, and (2) it will help to the authors to build the research hypotheses to test in the research. Here the second flaw, the paper does not present any hypothesis to test even though the authors have investigated the same topic before in almost a half-dozen species. Hypotheses to test are needed in a research paper. The third issue I found is the repetition of some of the data presented in the results section, and the lack of strategy (based on hypotheses) to determine when the data from all seeds collected and all treatment assayed is pulled together, when is presented for individual trees (seed lots), or when is presented for all combinations of seed lots and treatments. Also, the stats used to analyse data that follows a binomial distribution must be revised (particularly ANOVA and Duncan). And finally, in my opinion, a big flaw is the predictive model constructed to determine the methylation levels of a seed at specific germination percentages. For example, the development of this model (and the data used to build it) is not well explained in the methods. One can intuitively understand what is the origin of the data used for the correlation shown in Fig 7. But the way figure 8 has been constructed is completely unclear (particularly when the control methylation values varied across seed lots and drying conditions). In addition, the model do not show the variation or the confidence interval, and the predictive power of the model is unclear, as a probability of prediction based on the variability of the data is not provided. This lack of explanation on how the model has been built, how a sigmoidal trend is obtained in Fig 8 (if fig 7 shows a linear trend of the raw data), and the low power of the model may interfere in the conclusions reached by the authors in terms of the use of the methylation level as an early indicator of seed aging. The interpretation of the data in this respect is crucial for this paper based on the title presented. With independency of the flaws indicated I think the data set presented is novel, very interesting and has a great potential, and I encourage the authors to improve the manuscript around the issues indicated above: (1) a good background on DNA methylation in seed ageing, (2) a set of hypotheses to test, (3) a proper and non-repetitive way to present data taking into consideration all variables (seed lot, drying level and temperature), and (4) a solid and well explained predictive/descriptive method to confirm why DNA methylation levels are an early indicator of seed aging. Other minor comments on the manuscript are also included.
Our response: We have acknowledged Reviewer’s suggestion regarding introduction paragraph, stating the clearer hypothesis, presenting data, and solid explanation of constructed model for prediction the germination decline. Substantial changes have been made in the manuscript, therefore please check the manuscript with tracked changes, were all introduced changes are marked.
Point-to-point response to Reviewer’s comments.
Comment: Title: The stored material in which DNA methylation is an early indicator of ageing are the seeds, not the “exceptional” plant species used for the experiments (Populus nigra). I think it would be more accurate if the title is edited a bit, and reads as: “DNA methylation as an early indicator of aging during the storage of the seeds of the “exceptional” species Populus nigra L.” or “DNA methylation as an early indicator of aging in stored seeds of the “exceptional” species Populus nigra L.”
Our response: Corrected. Current version of title: DNA methylation as an early indicator of aging in stored seeds of “exceptional” species Populus nigra L.
Comment: Abstract: Authors are using the scientific name of the species through the whole abstract, except in line 28 “Once poplar seeds reached a 10% reduction in the level of m5C”. For consistency and to avoid confusion to the reader (who may not be familiar with the common name of the species and this has not been defined in the abstract) I would use the term “P. nigra” instead of “poplar” in this particular sentence of the abstract: “Once P. nigra seeds reached a 10% reduction in the level of m5C”.
Our response: Corrected. Current version: “Once P. nigra seeds reached an 8-10% reduction in the level of 5mC, a substantial decrease in germination occurred.”
Comment: Line 38, where it says “conversation” should say “conservation”.
Our response: Corrected.
Comment: Lines 49-51 “seed banking is based on desiccating seeds at 15% of relative humidity (RH) to a specific moisture content (MC) followed by storage at -20 °C”. In this sentence, if authors are talking about the RH for desiccation before storage, I recommend following the FAO standards, that indicate that drying can be done at a range of RHs: 10-25 % (and a range of temperatures: 5–20 °C). The 15% RH indicated for drying in seed banking, although correct, is not the standard in all seed banks (see, e.g., seed banks in the CPC network), and the sentence should be more inclusive of other drying conditions supported by FAO standards. However, if authors want to indicate that the RH at the storage temperature should be 15%, then this would ok as per FAO standards, but the sentence should be edited accordingly.
Our response: Corrected. Current version: “One of the main reasons underlying this shortfall is that seed banking is based on desiccating seeds at 10-25% of relative humidity (RH) to a specific moisture content (MC) followed by storage at -20 °C. This protocol is efficient for long-term conservation of orthodox seeds [5,6].”
Comment: Paragraph between lines 60-64. I fully agree with this paragraph, and I am glad the authors mention the issue of diversity of ageing mechanism and the use of just a few model species to study seed ageing. However, I would extend the discussion on this issue a bit more (maybe one more sentence here or in the discussion section). For example, ageing may be dependent on the physical-chemical characteristics of the cells in the seeds (so called dry architecture in Ballesteros D, Pritchard HW, Walters C (2020). Dry architecture: towards the understanding of the variation of longevity in desiccation-tolerant germplasm. Seed Science Research 30, 142–155. https://doi.org/10.1017/S0960258520000239). This is particularly relevant for this paper, as for example Arabidopsis seeds (the model species mentioned by the authors here) are oily seeds in which ageing may be related to the oxidation or/and crystallization/melting issues induced by/ produced in the storage lipids, and Populus nigra seeds contain developed chloroplasts in the mature seeds that may modulate their ageing during storage (see conclusions in Roqueiro G et al. 2010. Effects of photooxidation on membrane integrity in Salix nigra seeds. Annals of Botany 105: 1027–1034, doi:10.1093/aob/mcq067, and comments on ageing of chlorophyllous seeds in Ballesteros et al., 2020). Could the authors somehow introduce this diversity of ageing mechanisms depending on the physical-chemical properties of their cells (or dry architecture) here? This is also relevant for the next paragraph when authors talk about ageing mechanism studied in P. nigra seeds. There, the potential damaging effect of the chloroplasts in the dry cells of the seeds has not been introduced. This is a suggestion for the edition of the text (edition in red) in case the authors want to take my comment/suggestion into consideration: “Fundamental knowledge about the progressive loss of seed viability during storage is based on studies conducted in crops and model species (Arabidopsis thaliana L.) which produce orthodox seeds of relatively long lifespan [9–12]. In the case of short-lived seeds, however, our knowledge of the factors determining seed viability is insufficient, which prevents the establishment of effective methods for long-term storage. In this respect, it has recently been suggested that ageing mechanisms may be different in seeds of different species based on their cell’s “dry architecture”, particularly in relation to seeds with short lifespan [Ballesteros et al., 2020]. For example, P. nigra seeds have very low storage lipids and contain developed chloroplasts in their dry mature cells, which may be the main source of oxidative stress inducing the fast ageing in these seeds [Roqueiro et al., 2010; Ballesteros et al., 2020]. On the other hand, A. thaliana seeds (one of the model species mentioned above), do not contain chloroplasts and have relatively high amounts of storage lipids that, instead of the chloroplast, may be more involved in the slower ageing of the seeds in this and other oily species [Ballesteros et al., 2020; Zinsmeister et al., 2020; Gerna et al., 2022]”. Zinsmeister, J., Leprince, O., & Buitink, J. (2020). Molecular and environmental factors regulating seed longevity. Biochemical Journal, 477(2), 305-323. Gerna, D., Ballesteros, D., Arc, E., Stöggl, W., Seal, C. E., Marami-Zonouz, N., ... & Roach, T. (2022). Does oxygen affect ageing mechanisms of Pinus densiflora seeds? A matter of cytoplasmic physical state. Journal of Experimental Botany, 73(8), 2631-2649.
Our response: We strongly appreciate that comment. The suggested corrections and introduction of additional text were made. Please see the manuscript with tracked changes.
Comment: Line 65: “Populus nigra L. (black poplar) is one of the most endangered tree species in WHERE? [13]” The place where P. nigra is one of the most endangered tree species is not indicated. It is the World? Europe? France? Poland?
Our response: Corrected. Current version: “Except potential differences in aging mechanisms between P. nigra and A. thaliana, black poplar seeds have been chosen for the research as this species is one of the most endangered tree species in Europe [17]. In Poland, this species is not legally protected as a species, but most populations are part of the Natura 2000 conservation program that prevents devastation of ecologically valuable ecosystems. However, a progressive reduction in black poplar population size is still being observed in Polish river landscapes [18].”
Comment: Lines 76-80: If not detailed above as suggested, I would add to this enumeration of physiological and biochemical measures of ageing in P. nigra seeds, the potential negative role of the chloroplasts that are part of the dry mature seeds stored. For example (editions or additions in red): “The physiological and biochemical changes of P. nigra seeds in relation to ageing have been the subject of several studies [13,18–23]. Specifically, these studies have demonstrated that storage of black poplar seeds results in increased oxidative stress [22], large changes in protein abundance [23], and a reduced proportion of roots with absorptive function [21]. Interestingly, many of these changes could be induced by an exacerbated oxidative stress mediated by the chloroplasts found in the dry mature seeds of this species [Roqueiro et al., 2010; Ballesteros et al., 2020]”
Our response: Corrected according to Reviewer’s suggestions. Please see manuscript with tracked changes.
Comment: Paragraph between lines 83-87. DNA methylation in seeds during ageing is the main topic of this paper, but the background on this topic has not been sufficiently introduced. A reader not familiar with the topic will need this information to understand the novelty of the research around the topic. Also, authors must introduce the state of the art in the topic “DNA methylation in seeds during ageing” so proper hypothesis for the research can be established. This is particularly relevant when the authors have investigated this topic previously in diverse species: see references cited in this manuscript [26, 27, 28, 36, 38]. In addition to their own research, I recommend adding information from the recent research made by other authors in this topic, like Mira et al (Mira, S., Pirredda, M., Martín-Sánchez, M., Marchessi, J. E., & Martín, C. (2020). DNA methylation and integrity in aged seeds and regenerated plants. Seed Science Research, 30(2), 92-100), Pirreda et al (Pirredda, M., González-Benito, M. E., Martín, C., & Mira, S. (2020). Genetic and epigenetic stability in rye seeds under different storage conditions: Ageing and oxygen effect. Plants, 9(3), 393), and Yalamalle et al (Yalamalle, V. R., Ithape, D. M., Kumar, A., Bhagat, K., Ghosh, S., & Singh, M. (2020). Seed treatment with 5-azacytidine reduces ageing-induced damage in onion seeds. Seed Science and Technology, 48(3), 407-412.).
Our response: Corrected according to Reviewer’s suggestions. Please see manuscript with tracked changes.
Comment: Lines 88-90: Authors state that “The objective of the present research was to determine if changes in global m5C levels affect or are correlated with the viability of short-lived, intermediate seeds produced by exceptional species.” However, based in previous findings of the authors and the other research on seed DNA methylation presented above, this objective is a bit trivial. Authors must set (at least) one novel hypothesis to test, particularly when they have investigated this topic before in diverse orthodox, intermediate, and recalcitrant seeded species [26, 27, 28, 36, 38] and there is recent research published in diverse orthodox seeded species [Mira et al., 2020; Pirredda et al., 2020, Yalamalle et al., 2020]. What is new in this research? For example, authors use dry vs moist storage at different low temperatures while Mira et al 2020 and Pirredda et al 2020 did their research on high temperature and moisture conditions. Or which previous unresolved questions are authors using to build up this research? For example, changes in DNA methylation during ageing are known, but their value as an early indicator of seed aging has not been proved. If hypotheses are not presented, the paper seems just the repetition of a method but in a different species. Authors must introduce an advance to this topic previously investigated in half-dozen species. Maybe a way to start is to transform the objective of the first sentence and the overall goal in the last sentence into proper hypotheses to test: “We hypothesize that changes in global m5C levels will be correlated with the viability of short-lived, intermediate seeds produced by the exceptional species P. nigra during dry and cold storage. In addition, we hypothesize that changes in DNA methylation (m5C) will be observed prior to the main reduction in seed viability”. But based on the author’s experience in this topic, I am sure they will have plenty of hypotheses they aimed to test with this research.
Our response: Substantial changes have been made in the manuscript, as some fragments have been rewritten or added. We acknowledged previous work made by other groups regarding the impact of aging on DNA methylation. The novelty of our current research was therefore described in the introduction. We have also taken into deeper consideration the relevance of seed moisture content for DNA methylation analysis. For this suggestion we are sincerely grateful, as we are certain that it added a new layer of information to our manuscript and strengthened our final statement about 5mC as a universal marker independent on an initial viability and a seed moisture content. As many changes have been made in entire manuscript, please see the manuscript with tracked changes.
Comment: Lines 96-98: Seeds were collected from three different trees in the same location near Czeszewo (52°8’N, 17°30’E), although separated 3 km one to each other. Was each seed collection (representing one single tree) considered as a “seed lot” for the data analyses? Please specify in the methods if this was the case. However, to me, these three trees seem like the remnants of a previously larger and now fragmented population. Is this Ok? Or are they three trees in a large population formed by hundreds of trees? Or three trees from three different populations? Or are they the single remnants of three different “near extinct” populations with just one or a few trees? Please clarify in the methods. This is important to consider the way to analyse the data. For example, if the three trees belong to a single population, why the seeds from the three trees were not pulled together and the three-tree collection was considered as a single collection? On the other hand, if authors wanted to study inter-tree differences within a population (maybe small and highly fragmented), this should be stated in the introduction, when setting the objectives/hypotheses, otherwise the reasons to treat the seed collection of each tree separately makes no sense to me.
Our response: We have used the term “seed lot” to describe not only seeds from one maternal tree, but also to underly their initial viability as the term “seed lot” usually describes seeds of one germinability. We have decided to keep this terminology. After discussion with the researchers from Institute of Dendrology, Polish Academy of Sciences, we assume that this is a one, aging population, however the genetic analyses of those trees are ongoing, therefore until final results are obtained no ultimate statements about those trees can be made.
Current version of the text: “Mature seeds of black poplar (Populus nigra L.) were collected from three different 50- to 70-year-old trees growing in one population growing near Czeszewo (52°8’N, 17°30’E) that were physically separated from each other by a distance of at least 3 km. Seeds were collected in the same manner as described in [25].”
Comment: Line 108: What is a “Jacobsen apparatus”? Please provide trademark and model and if possible, reference stating its use for seed germination previously, as not all readers may be familiar with this apparatus.
Our response: Corrected. Current version: “Germination tests were conducted using 50 seeds placed on moist filter paper (70 mm in diameter) in a Jacobsen apparatus type 5101 (Rumed, Laatzen, Germany) covered with a plastic lid.”
Comment: Line 145: In the sentence “The effect of desiccation and storage on seed germination”, does “storage” relate to “temperature”? If so it should read as: “The effect of desiccation and storage temperature on seed germination”. Same in line 147 “impact of desiccation and storage temperature on the level of DNA methylation”.
Our response: Corrected.
Comment: Lines 145-146: Here it is indicated that germination data for the GLM was treated as data following a binomial distribution. I suppose that this is due to the fact that each seed placed to germinate was considered as a “sample” and data was treated in a binomial way as success (germination) and no success (no germination) to build a proportion. Is that ok? But authors mention in lines 108-110 that 50 seeds were germinated in 4 technical replicates. Does this mean that the data from the 200 seeds (in 4 technical replicates) used for each conditions tested were pulled together for the data analysis in the GLM? Please specify the treatment of the data and replicates.
Lines 151-155: In line 145-146 is indicated that germination data was considered to follow a binomial distribution. However, lines 151-155 indicate that different tests were used to determine differences in the mean of germination across different drying and storage conditions. The tests used as per figures 1, 3 and 5 ANOVA and Duncan tests. However, these tests are suitable for data that follow a normal distribution, and not for the germination data that do not follow a normal distribution (indeed they were considered to follow abinomial distribution). Here there is a methodological problem. If authors want to determine differences in the mean of germination across different drying and storage conditions using ANOVA and Duncan test the raw germination % data must be transformed to approach the data into a normal distribution (e.g., by the arcsine transformation). Otherwise, proper “binomial tests” to account for differences in the proportion of germination across different drying and storage conditions must be used (e.g., Z test, Chi-square, Fischer,… here some indications for the Z-test: http://www.sthda.com/english/wiki/two-proportions-z-test-in-r )
Lines 161-164: The way this prediction was done needs to be better explained. For example, what was the original data used to construct/model the relation between germination and m5C level (table 2)? Was the data shown in the correlation expressed in Fig. 7? This means that data from all three seed lots and drying conditions was pulled together, isn’t it? Are not the authors curious about how the desiccation levels affect the correlation of methylation levels vs germination%? I am, though, as ageing mechanisms may be different in the non-dried and the dried seeds. BTW, when modeling this relation, what was the link used in the GLM? Logistic? Probit? Linear? It looks linear from data in Fig. 7, but this needs to be well defined in the methods. Finally, what data was used for constructing Figure 8? It is not clear in any place. For example, what was the reference (control) value of methylation used to calculate the relative percentage of change in m5C level in relation to control? If this value changed depending on the seed lot used and the desiccation level (Fig. 4), how was this done? Why figure 8 lacks variation in the data set? Supplementary information with all these calculations would be helpful to evaluate the model constructed.
Our response: We agree with the Reviewer that germination data follows a binomial distribution. Therefore, linear models with ANOVA and post hoc tests cannot be applied as one of the main assumptions of the ANOVA is that each factor must have a normal population distribution. In case of germination capacity data, this assumption is violated and excludes the possibility to use a linear model and ANOVA. Consequently, we did not use the linear model and ANOVA to analyze our germination data, instead, we applied generalized linear model with binominal distribution which is dedicated to data with binominal distribution. The Reviewer’s concerns regarding applied statistical method could result from our mistake in Section Materials and Methods. Instead of “general linear model (GLM) with a binomial distribution” it should be stated “generalized linear model (GLM) with a binomial distribution “ as such analysis was applied. This was a simple misspelling. We are very grateful to the Reviewer for pointing it out. Thanks to that we were able to correct our mistake. The “glm” function in R, which was applied by us, corresponds to Generalized Linear Model. As the general linear model requires that the response of variable follows the normal distribution whilst the generalized linear model is an extension of the general linear model that allows the specification of models whose response of variable follows different distributions (for instance binominal distribution).
We have corrected the sentence and we also added information about link function used in GLM. Current version: The effect of desiccation and temperature on seed germination was separately evaluated using a generalized linear model (GLM) with a binomial distribution and Logit link function [39–41].
We would like also to clarify why we had used GLM with binominal distribution instead of arc sin transformation what was suggested by the Reviewer. We agree that for many decades arcsine transformation was used as variance-stabilizing transformation for percentage data. Authors used this transformation in their previous publications as well. However, in last decade strong concerns have arisen about applying arcsin to transformation of percentage data. For instance, in the publication from Warton and Hui 2011 “The arcsine is asinine: the analysis of proportions in ecology”, authors stated: “The arcsine square root transformation has long been standard procedure when analysing proportional data in ecology, with applications in data sets containing binomial and non-binomial response variables. Here, we argue that the arcsine transform should not be used in either circumstance.” Easy analysis of binominal distributed data could be conducted by generalized linear model (GLM) with a binomial distribution what is currently much more commonly used for analyses of germination data (over 119 publications on ISI Web of Science; database checked on 09.06.2022), with significant increase during last 5 years. Also, the advantages of using generalized linear model for germination data are discussed in Carvalho et al. 2018 or Amorim et al. 2021. Another reason supporting the current common application of the GLM or GLMM is the fact, that even though the mathematics of GLMM estimation are challenging (Jiang 2007), nowadays they can be easily performed in R. Summarizing, based on arguments mentioned above, we have a confidence that statistics applied by us which was generalized linear model (GLM) with a binomial distribution is optimal for seed germination test and consistent with the latest findings in biostatistics.
Amorim D. J., dos Santos A. R. P., da Piedade G. N.,de Faria R. Q., da Silva E. A. A., and Sartori M. M. P. The Use of the Generalized Linear Model to Assess the Speed and Uniformity of Germination of Corn and Soybean Seeds. Agronomy 2021, 11, 588.
Carvalho FJ, Janoni F, de Santana DG, de Araújo LB. (2018) Why analyze germination experiments using Generalized Linear Models? Journal of Seed Science, v.40, n.3, p.281-287, 2018
Warton DI, Hui FKC. (2011). The arcsine is asinine: the analysis of proportions in ecology. Ecology, 92(1), 3–10.
Jiang, J. 2007. Linear and generalized linear mixed models and their applications. Springer, New York, New York, USA.
Line 161-164
We agree with the Reviewer 2 that testing how desiccation levels affect the correlation between DNA methylation levels and germination % could be interesting. However, our main goal was to verify whether 5mC could be a universal early marker of asymptomatic ageing regardless MC of seed and their initial viability. We provided correlation of germination and DNA m5C for all seed lots at hydrated and desiccated MC in Table 1. The correlation results show that there is no substantial difference among correlation between DNA methylation and germination for each tested seed lot, including dried and hydrated seeds and seeds collected from different maternal trees. Indeed, in all cases corelation coefficients ranged between 0.694-0.87 and were highly significant at level p<0.001. Based on these results we were able to pull all data (data obtained from all seeds) together, as the main aim of our research was to show that DNA methylation level decline is related to germination decrease in aging seeds regardless temperature or moisture content of storage. However, seed MC remained at safe range with no observed desiccation -related decrease in germination. Therefore, based on all data we used p.dose function in R to construct the model of Δ%5mC / %of germination. As the Reviewer suggested to analyse data separately for moist and dried seeds, we added separate correlations for both batches in corrected Table 1. The correlation between DNA methylation and germination % for combined desiccated seeds from three seed lots was R=0.775 (p<0.0001) and for non-desiccated seeds R=0.822 (p<0.001). These two values were close to correlation coefficient for all seeds R=0.7965 (p<0.0001). Consequently, this result provides additional support to our statement, that the measurements of DNA methylation can be considered as universal marker of seeds aging as it is significantly correlated with germination both in dried and non-dried seeds. However, we totally agree with the Reviewer that the mechanism of aging is expected to be different in the non-dried and dried seeds, what is the goal for our next research.
Based on the results showing highly significant and comparable correlation between DNA methylation changes and germination % for all tested seed lots at both (high and low) MC, we decided to use all data to construct p.dose glm model with binominal distributions. The model that has been chosen by us was characterised by low AIC (Akaike information criterion which is an estimator of prediction error and thereby relative quality of statistical models for a given set of data) which was 73.605 and Null deviance was 100.375 on 119 degrees of freedom. What is important to state is that Pr(>|z|) value for the obtained model was very low, exactly 1.53e-08, so we stated it in the manuscript at p<0.0001, (N=120), (more data about model are in supplementary Table S1). Therefore, all parameters indicate that constructed p.dose model is of very high significance. We constructed similar models for non-desiccated seeds and desiccated seeds according to excellent suggestion made by Reviewer. Model 2 for non-desiccated seeds only, was characterized by N=60 and AIC: 41.184, Pr(>|z|) was 6.82e-05, Null deviance was 60.186 on 59 degrees of freedom. Model 3 for desiccated seeds only was characterized by N=60, AIC: 34.459, Pr(>|z|) 0.000261, Null deviance was 35.396 on 59 degrees of freedom (more data about models are in supplementary Table S1). So, for all three constructed models null deviances were close to/lower than degrees of freedom, what additionally proves that these models are well fitted.
All three models (for all seed, desiccated seeds, non-desiccated seeds) are represented on corrected figure 6. As control to calculate relative changes (Δ%) in DNA methylation level, we used data obtained from the model for germination at 95%. We decided to use calculated data instead of real control data from separate seed lots, because this approach allowed us to conclude with the general statement about changes in DNA methylation during aging of Populus nigra seeds what was the main aim of the presented research. Additionally, we have decided to use 95% of germination instead of 99% or 100%, because p.dose model does not allow to calculate values for 0 or 100% as all estimates close to 0% and 100% had significantly higher standard error of obtained values which reflects less accurate model for outliers data. Third reason supporting the use 95% germination as control data was the fact that median value for control non-stored seeds was close to 95%. To clarify our approach, the method how graph was constructed is added to supplementary data with calculated relative DNA methylation changes in three created models (Supplementary Table S2). Briefly, to construct the graph, for six arbitrary chosen germination levels (5%, 10%, 25%, 50%, 80%, 90%, 95%), DNA methylation level were calculated from the model with p.dose function. According to Reviewer’s suggestion we have decided to show data obtained by p.dose function on GLM model in corrected Table 1. We have decided to choose these particular levels of germination as some of them are interesting for gene banks (i.e. 80% germination). In the Table 1, thanks to well fitted model, DNA methylation level at all chosen germination points are characterised by small standard errors. After, the difference between DNA methylation at 95% of germination (considered as 100% of m5C) was used to calculate the relative change (Δ%) in DNA methylation using 5mC % for arbitrary chosen germination points (as percentage of total 100% m5C). The result (%Δ m5C vs. %germination) were plotted to obtain sigmoidal curve.
Calculation:
X= % 5mC at 95% germination calculated by p.dose from GLM model (control)
Y= % 5mC at one of the arbitrary chosen germinations % calculated by p.dose from GLM model.
Z= Y/X x 100%
Δ= 100%-z.
Calculated Δ was plotted on x axis.
We added the methodology of calculation of Δ%m5C as Supplementary Equation 1.
Models for non-desiccated (green) and desiccated (orange) seeds were constructed based on the same calculation method. Although there are some differences in the slopes (see Supplemetary Table 3), in all three cases (all seeds vs. dry seeds. vs. hydrated seeds) the shape of slopes is similar (Figure 6). Differences in slopes, probably resulting from different aging process ongoing in seeds at varied MC, are visible, however for all of them the initial asymptomatic stage of germination ranges between 8-10% of Δ%5mC. This data additional support our conclusion that DNA methylation could be universal marker of initial stage of seed aging. Nevertheless, this aspect of research is very interesting and we are going to proceed with experiments explaining the differences in demethylation process.
To clarify the approach how the models were constructed and how significant they are we have decided to add more information in 1) material and methods chapter, in 2) results paragraph and 3) supplementary section. Because models were constructed on p.dose values, we decided also to remove points from the graph to make it less confusing why it does not have variation in the data set. We strongly believe that those changes made the information clearer and easier to understand. Please see the manuscript with tracked changes.
Comment: Line 167: P. nigra needs to be in italics
Our response: Corrected.
Comment: Line 168: Each seed lot are seeds from a single tree, then I am not sure if the term “seed lot” is correct or it should just say “tree”.
Our response: We have decided to keep the term seed lot as decribed above.
Comment: Lines 169-170, In the sentence “Control seeds did, however, exhibit a difference in initial seed germination capacity (Figure 1)”, are the authors stating that the seeds from each tree (seed lot) showed a different germination % before drying? Maybe the sentence can be edited for clarity.
Our response: The initial germinability of seeds differed before desiccation as well as after. Thank to that were able to verify the hypothesis that the measurement of 5mC change as aging indicator is independent on initial viability. This is explained in the manuscript.
Comment: Line 176: Authors use diverse terms to refer to the same treatment: freshly collected seeds, control seeds, non-desiccated seeds, seed lot at a 14.5% of MC,… to avoid confusion and facilitate the reading and interpretation of results, please, be consistent with the way this treatment is referred along the text (results and discussion). The use of “control seeds” is not recommended as it is used with different meanings in different sections of the manuscript: e.g., control seeds are referred to non-desiccated seeds (line 169), but also to non-stored seeds (although they may have been desiccated: line 208)
Response: To make the information in the manuscript clearer to reader, the term “control” was exchanged into non-stored or non-desiccated seeds were relevant
Comment: Lines 187-213 (section 3.3): I think this section can be summarized and redundancies can be removed (particularly when data is shown in Fig 3), focusing the description of the results in the common patterns observed. The seeds from the three seed lots (trees) responded to the combination of moisture and temperature in the same way, although with some differences in % germination likely due to the different initial germination of each lot/tree. In all three seed lots, the absence of desiccation resulted in very low or nil germination at +10 and +3C, and often (2 out of 3 lots) a small reduction of germination after storage at -20C. On the other hand, desiccation resulted in high germination (similar to non-stored seeds) after storage at -20C and +3C (except for seed lot 1). Germination after storage at -10 was also low, but higher than that of non-desiccated seeds. Germination after LN did not show any significant decrease independently of the drying treatment. Interestingly germination after LN storage increased a bit in lot 1.
Lines 221-252 (section 3.4): Similarly to the comment made above, I think this section can be summarized, focusing the description of the results in the common patterns observed (which are in relation to the common patterns observed in section 3.3).
I am confused about sections 3.5 and 3.6, corresponding to figures 5 and 6. Why now data from figures 3 and 4 is presented again but without considering the desiccation level or seed provenance (lot/tree)? What is the value of presenting data in a different way if the message is the same but the desiccation component is lost? If the seed provenance changes the results observed, I would understand that data for each desiccation level and temperature combination is presented for “all seed collection (data from the three trees pulled)” vs “each tree harvested (the three seed lots used in the paper)”. However, this is not the case. Hence, I think sections 3.5 and 3.6 are redundant and should be removed.
Table 1 and Figure 7. Again, I do not understand why data is presented for all seed collection vs each single tree. Figure 7 is sufficiently self-explicatory for the correlation found. If the inter-tree differences were an objective of the paper and there were hypotheses to test in this regard, I understand data is presented in these two ways. But this is not the case based on the introduction of the paper.
Our response: The results chapter has been shortened, redundant information has been removed. Figures showing results for all seeds were not removed completely but moved to supplementary files, as on basis of those data the predictive models were constructed, therefore the figures are important for reader to control data processing and visualize the aging effect on data pulled together from three seed lot at high and low MC.
Comment: Line 295 and 297: P. nigra needs to be in italics
Our response: Corrected.
Comment: Table 2: why germination % does not include a measure of variance as indicated for the methylation level?
Our response: For germination percentage measure of variance are not provided as they are arbitrary chosen germination points. Explained above in the response to comments about material and methods section and used statistics.
Comment: Lines 296-301 and Figure 8: I am not convinced about the model presented here and the interpretations made on the value of the methylation levels to be used as an early indicator of seed aging. First of all, the methodological procedure to define the model is not well explained in the methods (already indicated above). In addition, the “relative decline in methylation levels” (X- axis in Fig. 8) is very confusing: where this relative value comes from? Was it individually calculated for the relative change in methylation at each storage temperature respect to non-stored seeds (control)? Then, did authors considered each combination of seed lot and moisture level (for which control values were different)? If so, why data points in figure 8 do not show any value of variation? For example, at relative decline in methylation “zero” (so the control values), germination % ranges from 75% (seed lot 1) to 93% (seed lot 2) and figure 8 shows a single point at 95%. A supplementary table showing all these calculations would help, and also the variation of the raw data source around the fitted sigmoidal line. To me, the value of figure 8 seem redundant (or artificial) when we have the raw correlation of the germination vs methylation and the estimated doses (methylation levels) for different germination % (which is extracted from the fitted correlation of Fig. 7 but using the dose.p function in R). With this correlation (figure 7) authors could estimate the probability of obtaining certain ranges of germination % based on the raw methylation levels. But I understand this raw correlation is not showing the early detection of seed ageing as methylation change, because the correlation between germination vs methylation (raw values in Fig 7) is linear (not sigmoidal). With a linear correlation is difficult to ascertain a level of methylation for which germination % changes but the methylation levels do not change (this would have been clear if the correlation is sigmoidal or composed by two linear trends). Fig. 8 is sigmoidal, I agree, but as said above, the source of the data used to make this figure is not clear. This figure is not built based on raw data or we would see larger dispersion of data points as in Fig 7. The construction of this model must be clarified and values of variation or probability need to be included.
Our response: Substantial corrections were made in the text describing data processing, calculation methods and construction of predictive models. All values showing significance of constructed models has been also provided. Please see the manuscript with tracked changes, as they have been introduced into several fragments of text, into tables and figures as well as into supplementary data. Nevertheless, we are sure that suggestions made by Reviewer helped us to improve the manuscript particularly in the fragments describing statistical methods and models.
Comment: A discussion of a research paper use to be written to discuss if the results showed in the results section support the hypotheses presented in the introduction. But since no hypotheses are presented in the introduction of this paper, the points to be discussed seem arbitrary. Nonetheless, the discussion raises some important issues that could conform some of the hypotheses that are lacking in the paper. For example, the importance of interspecific seed quality (from different populations or single trees/lots, as is the case here) on the initial methylation levels (lines 331-333) but how the relative changes in methylation during seed ageing are independent of these initial levels (lines 395-397). Or if there is an interaction in the levels of methylation (or their relative decrease) during storage at different temperatures based on the hydration levels (said in other words, does methylation change in the seeds in a different way when they are dry or wet?). In my opinion, the discussion will have to be rewritten based on the hypotheses presented.
Lines 385-402. This is a fundamental piece of the paper. However, to validate this interpretation, the model constructed (Fig. 8) need to be clarified and the measures of variation need to be included.
Our response: The discussion paragraph has been corrected and authors believe that now the final statement is much clearer and better justified by the improvement in description of results, clarification of statistical methods and data processing. Construction of three models for all seeds, desiccated and non-desiccated seeds, suggested by Reviewer, allowed to verify the hypothesis described in the introduction and supports our statement about 5mC as an early indicator of aging regardless initial viability and seed MC.
Reviewer 3 Report
In the submitted manuscript, Michalak et al. describes the relationship between global DNA methylation level and germination ratio of Populus nigra L. seeds. The authors showed that global DNA methylation was decreased during seed storage, and it associated with the reduction of germination. Further, the levels of DNA methylation decline were strongly linked with the storage temperature. It must be important for wide range of readers. In the meantime, there are some aspects that fall short as detailed below, these need to be addressed.
The authors represent DNA methylation as m5C. However, conventionally 5mC is used to represent 5-methyl cytosine, m5C indicates methylation for RNA. Thus, 5mC or methylation must be used for this manuscript.
Line 140-141. The calculation should be %=5mC/total C (5mC and C)x100.
The authors used moisture content percentage to represent samples in the whole manuscript. However I think to use the words non-desiccated (control) and desiccated seem easy to understand for readers.
Line 177 and 178. What does R1 mean?
Fig. 3 and 5, and results 3.3 and 3.5 should be combined, Fig. 4 and 6, and results 3.4 and 3.6 also.
Fig. 3b is not mentioned in the manuscript.
In my opinion, the description of results 3.3 and 3.4 seem repetitive, should be shortened considerably.
Fig. 4a. The DNA methylation level was significantly elevated at -196 compared with control. The authors should explain and discuss reason why.
Line 361. The R2 value should be corrected.
Line 372 and 401. In my opinion, these descriptions seem inconsistent, because active DNA demethylation machinery relies on base excision repair pathway.
Line 398. It is not clear for me what does the description “impair (epi)genetic stability” mean. Overall, the molecular mechanisms why DNA methylation associates with germination ratio are sill unclear. Possible working model should be discussed.
Author Response
Response to Reviewer 3
We would like to thank the Reviewer for the contribution to perfecting the manuscript. We have improved the manuscript according to the suggestions. We hope to get the Reviewer’s approval.
Comment: The authors represent DNA methylation as m5C. However, conventionally 5mC is used to represent 5-methyl cytosine, m5C indicates methylation for RNA. Thus, 5mC or methylation must be used for this manuscript.
Our response: The suggestion was acknowledged and the corrections were made.
Comment: Line 140-141. The calculation should be %=5mC/total C (5mC and C)x100.
Our response: Corrected.
Comment: The authors used moisture content percentage to represent samples in the whole manuscript. However I think to use the words non-desiccated (control) and desiccated seem easy to understand for readers.
Our response: In entire manuscript the corrections have been made and seed lots are described as non-desiccated or desiccated if the exact moisture content value is unnecessary at specific point.
Comment: Line 177 and 178. What does R1 mean?
Our response: R1 corrected to R
Comment: Fig. 3 and 5, and results 3.3 and 3.5 should be combined, Fig. 4 and 6, and results 3.4 and 3.6 also. Fig. 3b is not mentioned in the manuscript. In my opinion, the description of results 3.3 and 3.4 seem repetitive, should be shortened considerably.
Our response: The substantial changes have been made to make the results chapter more concise and easier to read. The text is shorter now but we decided to keep as many data as possible, as based on these data further calculations are made. We have moved the figures 5 and 6 to supplementary data files, as we still think that it is important for the manuscript to keep them, because the calculations of relative change in methylation is based on combined data from all seed lot that are presented on these figures.
Comment: Fig. 4a. The DNA methylation level was significantly elevated at -196 compared with control. The authors should explain and discuss reason why.
Our response: As we do not expect any enzymatic activity during cryostorage, we assume that that result is an unexpected sampling issue, as there were no similar observations in other samples.
Comment: Line 361. The R2 value should be corrected.
Our response: Corrected.
Comment: Line 372 and 401. In my opinion, these descriptions seem inconsistent, because active DNA demethylation machinery relies on base excision repair pathway.
Our response: We have corrected that fragment. Current version:
“It can be also assumed that the demethylation observed at higher temperatures is due to an active demethylation via base excision repair process (BER) process catalyzed by DNA demethylases when 5mC is directly removed [50] or thymine-DNA glycosylase (TDG) when the potential oxidation products of 5mC (5-hydroxymethylcytosine, 5-formylcytosine and 5-carboxycytosine) are recognized and cleaved [53]. Regarding the latter, an increase in ROS (O2-● and H2O2) was observed in P. nigra seeds after one year of storage at +3 °C and -20 °C [27].”
Comment: Line 398. It is not clear for me what does the description “impair (epi)genetic stability” mean. Overall, the molecular mechanisms why DNA methylation associates with germination ratio are still unclear. Possible working model should be discussed.
The indicated fragment has been rewritten to make it more concise and easier to reader. The focus was gained on the known relation between DNA repair during imbibition and germination.
Our response: Current version: “It seems that the integrity of genomic DNA cannot be maintained at this stage and that efficient and accurate DNA repair is no longer possible [57]. Indeed, seeds have developed a repair system that allows them to minimize damage and repair biological molecules and cellular structures during imbibition, therefore the ability of seeds to repair itself are closely linked to their germination capacity [58]. “
Round 2
Reviewer 2 Report
I must acknowledge the authors as they have done an excellent work responding all queries and suggestions, and the manuscript is greatly improved. I just have some minor comments for the authors to respond before publication. Otherwise, an excellent paper, congratulations.
Minor comments:
Lines 170-174: I just want to emphasize/clarify that I was in fully agreement with the choice of the GLM for the analysis of the germination data made by the authors. This section didn´t need to be clarified but the small clarifications made on “generalized” vs “general” and the addition of the link used are great, thanks. My concerns were related to the tests used to determine significant differences between “mean values” after drying or after the storage of seeds at diverse temperatures, and for that reason I mentioned the issue of the transformation of the data (although it is not the ideal scenario as perfectly described by the authors in heir response). GLM doesn’t use “mean values” and their variance but fit data to different type of regressions based on a particular link and study the deviation to that model. So, my concerns were (and are): (1) how was the data of the germination treated? In other words, if authors germinated 50 seeds in 4 replicates (so a total of 200 seeds) were the counts for all 200 seeds pulled together to build the GLM? Or they used another method that accounted for the differences among technical/biological replicates? (2) why the results of the GLM model run have not been used to determine/explain the significance of drying and storage temperature on germination in the main text of the manuscript? Instead, the authors have used another set of tests based on the significance of Wilcoxon–Mann–Whitney tests for drying (Fig 1) and the significance of ANOVA and Duncan for storage temperature in the two different hydration levels (Fig 3). If a GLM was run to test for significant differences in germination after drying and after storage at different temperatures, should not the results of the model used inform about these differences? Then, why a Wilcoxon–Mann–Whitney test was run? Or ANOVA/Duncan? I can somehow understand the use of a “binomial” ANOVA and a post-hoc for temperatures but the way this was used need to be explained (as it should have been corrected for non-parametric data, see comment below).
Lines 178-180: In addition to the comment above, is the Wilcoxon–Mann–Whitney test the most convenient for data that follows a binomial distribution? Even though this is a test for non-normal distributed data, it is not recommended for counts (like data in binomial distributions). After a GLM, orthogonal contrasts or pairwise z-tests for proportions tend to be the test recommended instead (this is also suggested by Carvalho et al 2018, reference used by the authors).
Lines 181-183: I agree with the use of ANOVA after a GLM, but authors should specify that they calculated p-values based on the results of the GLM model and using a chi-squared test and chisquared distribution rather than the F-distribution which is reserved for parametric statistics. If this was not considered, then the stats should be re-run properly. Regarding the use of Duncan’s multiple range test as post hoc, same comment made above: even though this is a test for non-normal distributed data, it is not recommended for counts (like data in binomial distributions). Pairwise z-tests for proportions tend to be the test recommended instead.
Lines 187-200 + Fig. 5 + Table 2 + Table S1 + lines 323-331: I really appreciate for the explanation on how the model was calculated but still have some doubts. Apologies for being so insistent on this aspect but I have found this part of the paper difficult to follow. As far as I have understood after reading the whole paper, firstly, the correlation between germination % and global 5mC level was determined (by Pearson correlation). This correlation is represented in fig 5 and table 1 for all seeds, just for desiccated and just for non-desiccated seeds. Because there was a significant correlation, then these correlations were modelled in a GLM (Table S1) and using dose.p the precise values of global 5mC level for diverse levels of germination % were represented (Table 2). Finally, assuming the 95% germination level as the maximum global 5mC level (and using those values in table 4), the percentage of decline in global 5mC level for the diverse germination % shown in table 4 were calculated (Table S2) and plotted against the % germination (Fig. 6). Is that correct? If so, please edit sentence in lines 187-1914 to avoid confusion, as the current text “The prediction of relative change (Δ%) in 5mC percentage in P. nigra seeds leading to initiation of seed viability decline during storage was modelled using the glm function with a binomial distribution available in R” is not completely accurate. The GLM was used to model the correlation of the global 5mC level (not the relative change (Δ%) in 5mC percentage) against germination %. When this small detail is corrected, then it should be ok to say “The level of 5mC at 90%, 80%, 50%, 25%, 10% and 5% of germination was calculated using the dose.p function on the basis of constructed models.” I think it will help to the reader that the logical order in which the predictive model was constructed is well detailed in the methods: (1) correlation by Pearson, (2) GLM model for global 5mC level vs germination%, (3) determination of global 5mC level for diverse germination % using dose.p function, (4) calculation of relative change (Δ%) in 5mC percentage for the various germination % in (3), (5) representation of relative change (Δ%) in 5mC percentage vs germination %.
Figure 1. Why statistical differences are shown after a Wilcoxon–Mann–Whitney test? No GLM for these samples?
Line 335: Table 2: The superscript (1) in “Predicted DNA methylation level (%)1” does not refer to any foot note in the table. Add foot note or eliminate the superscript.
Line 336: Table 2: “* Viability critical for storage at gene banks. Data significant at the level of p<0.0001.” I understand that the * symbol in the 80% germination line refers to the % of viability considered by FAO as the lower % of viability that a collection must have to be considered “healthy” or “non-aged” in conventional gene banks. But what means that these values are also “significant at the level of p<0.0001”? The other values in the table were not significant? Or this significance is referred to other things (maybe to superscript 1)? Please clarify.
Author Response
We would like to sincerely thank the Reviewer for all submitted comments. We are having the impression that our discussion with the Reviewer not only improved our manuscript but also expanded our knowledge about statistical analyses regarding seed research. We strongly appreciate the effort made by the Reviewer to help us to clarify our findings.
Point-by-point response.
Comment: I must acknowledge the authors as they have done an excellent work responding all queries and suggestions, and the manuscript is greatly improved. I just have some minor comments for the authors to respond before publication. Otherwise, an excellent paper, congratulations.
Minor comments:
Lines 170-174: I just want to emphasize/clarify that I was in fully agreement with the choice of the GLM for the analysis of the germination data made by the authors. This section didn´t need to be clarified but the small clarifications made on “generalized” vs “general” and the addition of the link used are great, thanks. My concerns were related to the tests used to determine significant differences between “mean values” after drying or after the storage of seeds at diverse temperatures, and for that reason I mentioned the issue of the transformation of the data (although it is not the ideal scenario as perfectly described by the authors in heir response). GLM doesn’t use “mean values” and their variance but fit data to different type of regressions based on a particular link and study the deviation to that model. So, my concerns were (and are): (1) how was the data of the germination treated? In other words, if authors germinated 50 seeds in 4 replicates (so a total of 200 seeds) were the counts for all 200 seeds pulled together to build the GLM? Or they used another method that accounted for the differences among technical/biological replicates? (2) why the results of the GLM model run have not been used to determine/explain the significance of drying and storage temperature on germination in the main text of the manuscript? Instead, the authors have used another set of tests based on the significance of Wilcoxon–Mann–Whitney tests for drying (Fig 1) and the significance of ANOVA and Duncan for storage temperature in the two different hydration levels (Fig 3). If a GLM was run to test for significant differences in germination after drying and after storage at different temperatures, should not the results of the model used inform about these differences? Then, why a Wilcoxon–Mann–Whitney test was run? Or ANOVA/Duncan? I can somehow understand the use of a “binomial” ANOVA and a post-hoc for temperatures but the way this was used need to be explained (as it should have been corrected for non-parametric data, see comment below).
Response: We are aware that there is no one and accepted “gold standard” for statistical analysis of seed germination data and there is ongoing scientific debate regarding the most appropriate method for analysis of seed performance. Significantly, there are previously published articles related to this topic only. We are aware that currently there are at least four statistical approaches in analysing seed germination a) applying Bayesian statistics, b) transforming germination percentage data and conducting ANOVA on transformed data, c) applying non-parametric tests such as Kruskal–Wallis test or Wilcoxon–Mann–Whitney test, and finally, d) using GLM with binomial distribution followed by ANOVA and post hoc test. Recommended by Reviewer 2 pairwise z-tests for proportions are not used for germination data. At least it is not commonly used, as we were not able to find any publication on ISI Web of Knowledge in which z-test was applied to analyse germination data. Therefore, we do not judge that this is not an optimal method, but we have decided to keep our current statistic approach testing germination after desiccation (Figure 1). We are very grateful to Reviewer 2 for pointing out that there exists another method of statistical analysis, which, as we are aware, is commonly applied for big data as i.e. biomed data or gene expression data. However, as it has not been used in seed data restricted in counts, we have had too little evidence that it could be easily and correctly applied for our results. We have decided to use statistical analysis which is easy to follow and understand by other groups as it is widely used in seed research. However, we will keep our eye on pairwise z-tests for proportions. Therefore, we greatly appreciate the discussion with Reviewer about this topic.
Comment: (1) how was the data of the germination treated? In other words, if authors germinated 50 seeds in 4 replicates (so a total of 200 seeds) were the counts for all 200 seeds pulled together to build the GLM? Or they used another method that accounted for the differences among technical/biological replicates?
Response: As it has been stated in the manuscript, GLM models were constructed for analysis of germination of three seed lots at two moisture content separately (presented in Figure 3). Clarifying, there were six separate GLM models constructed followed by ANOVA and post hoc tests. An additional GLM model was constructed for all combined germination data (Figure 5). We used 50 seeds per biological replication, and each replication was incorporated separately into the model.
Comment: Lines 178-180: In addition to the comment above, is the Wilcoxon–Mann–Whitney test the most convenient for data that follows a binomial distribution? Even though this is a test for non-normal distributed data, it is not recommended for counts (like data in binomial distributions). After a GLM, orthogonal contrasts or pairwise z-tests for proportions tend to be the test recommended instead (this is also suggested by Carvalho et al 2018, reference used by the authors).
Response: As it has been mentioned above, despite the fact that there is ongoing scientific debate about the most appropriate way to perform statistical analysis of seed germination data, there is a lack of information about applying pairwise z-tests for proportions for this purpose. Pairwise z-tests for proportions are well established with analysis of big data in biomed studies. However, computing the correct Z-test is also not easy as it was stated i.e. by Chen and Nadaraja in the paper entitled: “On the optimally weighted z-test for combining probabilities from independent studies”. However, we do not want to judge whether this test is not appropriate for the analysis of germination. However, so far it has been not commonly used, actually we have failed in finding any publications showing Z-test in germination data.
Unfortunately, we were not able to conduct orthogonal contrasts which was mentioned in Carvalho et al 2018, as orthogonal contrasts for analysis of variance are independent linear comparisons between the groups of a factor with at least three fixed levels, and we have two levels of moisture content - desiccated vs. non-desiccated seeds. For the same reason, we were not able to apply GLM and ANOVA as one-way analysis of variance (ANOVA) is used to determine whether there are any statistically significant differences between the means of three or more independent (unrelated) groups.
We have found no publications related to the germination of seeds showing a t-test on GLMs data, so we have decided to apply a non-parametric test (Wilcoxon–Mann–Whitney) instead. As Reviewer 2 suggested based on the summary of GLM we observed that there is no significant differences between treatments. Pr(>|z|) 0.982 for Seed lot No. 1, Pr(>|z|) 0.9763 for Seed lot No. 2, and Pr(>|z|) 0.771 for Seed lot No. 3. Pr(>|z|) in binomial distribution corresponds to p-value and it is interpreted in a similar way. However, we did not find in the literature that based only on GLM the significant difference between treatments can be assessed, therefore, we have decided to present the data analyzed with the non-parametric Wilcoxon–Mann–Whitney test.
Additionally, we also observed that DNA methylation data are not normally distributed in desiccation-oriented experiments, therefore, we have decided to perform a non-parametric Wilcoxon–Mann–Whitney test in both cases (Figures 1 and 2).
Comment: Lines 181-183: I agree with the use of ANOVA after a GLM, but authors should specify that they calculated p-values based on the results of the GLM model and using a chi-squared test and chisquared distribution rather than the F-distribution which is reserved for parametric statistics. If this was not considered, then the stats should be re-run properly. Regarding the use of Duncan’s multiple range test as post hoc, same comment made above: even though this is a test for non-normal distributed data, it is not recommended for counts (like data in binomial distributions). Pairwise z-tests for proportions tend to be the test recommended instead.
Response: As has been described in the manuscript, we have used GLM. In GLM p-values are calculated using chi-squared distribution. This information has been added into materials and methods section.
Comment: Lines 187-200 + Fig. 5 + Table 2 + Table S1 + lines 323-331: I really appreciate for the explanation on how the model was calculated but still have some doubts. Apologies for being so insistent on this aspect but I have found this part of the paper difficult to follow. As far as I have understood after reading the whole paper, firstly, the correlation between germination % and global 5mC level was determined (by Pearson correlation). This correlation is represented in fig 5 and table 1 for all seeds, just for desiccated and just for non-desiccated seeds. Because there was a significant correlation, then these correlations were modelled in a GLM (Table S1) and using dose.p the precise values of global 5mC level for diverse levels of germination % were represented (Table 2). Finally, assuming the 95% germination level as the maximum global 5mC level (and using those values in table 4), the percentage of decline in global 5mC level for the diverse germination % shown in table 4 were calculated (Table S2) and plotted against the % germination (Fig. 6). Is that correct? If so, please edit sentence in lines 187-1914 to avoid confusion, as the current text “The prediction of relative change (Δ%) in 5mC percentage in P. nigra seeds leading to initiation of seed viability decline during storage was modelled using the glm function with a binomial distribution available in R” is not completely accurate. The GLM was used to model the correlation of the global 5mC level (not the relative change (Δ%) in 5mC percentage) against germination %. When this small detail is corrected, then it should be ok to say “The level of 5mC at 90%, 80%, 50%, 25%, 10% and 5% of germination was calculated using the dose.p function on the basis of constructed models.” I think it will help to the reader that the logical order in which the predictive model was constructed is well detailed in the methods: (1) correlation by Pearson, (2) GLM model for global 5mC level vs germination%, (3) determination of global 5mC level for diverse germination % using dose.p function, (4) calculation of relative change (Δ%) in 5mC percentage for the various germination % in (3), (5) representation of relative change (Δ%) in 5mC percentage vs germination %.
Response: The logical order proposed by Reviewer 2 is correct. However, we would like to clarify that in GLM global 5mC level vs. germination % were modelled, not results from Pearson correlation. Therefore, we added such information to the Material and Method Line 189-190
Comment: Figure 1. Why statistical differences are shown after a Wilcoxon–Mann–Whitney test? No GLM for these samples?
Response: Please see our detailed response above. Briefly, we have decided to use a non-parametric test as we did not find any confirmation in the literature that t-tests can be used on GLM data. However, we provided the significance level of GLM. To avoid problems with interpretation of our data, we have decided to use a non-parametric test that could be conducted on non-normal distributed data, and Wilcoxon–Mann–Whitney is commonly used to find differences between two treatments in germination data.
Comment: Line 335: Table 2: The superscript (1) in “Predicted DNA methylation level (%)1” does not refer to any foot note in the table. Add foot note or eliminate the superscript.
Response: Corrected.
Comment: Line 336: Table 2: “* Viability critical for storage at gene banks. Data significant at the level of p<0.0001.” I understand that the * symbol in the 80% germination line refers to the % of viability considered by FAO as the lower % of viability that a collection must have to be considered “healthy” or “non-aged” in conventional gene banks. But what means that these values are also “significant at the level of p<0.0001”? The other values in the table were not significant? Or this significance is referred to other things (maybe to superscript 1)? Please clarify.
Response: Indeed, the significance level refers to superscript 1. The mistake has been corrected.
Reviewer 3 Report
Most of my concerns are improved in the revised version of the manuscript. However there are still several aspects that I hope the authors to improve as follows.
The authors moved former Fig. 5 and 6 as Supplementary Fig 1 and 2 in the revised version. I think these results are very important and should be included in the main body of the manuscript. I would like to recommend the authors that Sup Fig. 1 and 2 should be added as Fig. 3g and 4g, or Fig. 5a and 5b, respectively.
It is clear that dynamic alteration of DNA methylation occurs with seed drying and aging. I would like to suggest to discuss (or to mention) whether these dynamic DNA methylation changes occur in embryonic tissues and/or endosperm.
Minor points
Line 325. The R value should be corrected.
Line 436. Tab. 1 should be Table 1.
Line 479. “base excision repair process (BER) process” should be “base excision repair (BER) process”.
Author Response
We would like to sincerely thank the Reviewer for all submitted comments and suggestions.
Point-by-point response.
Comment: Most of my concerns are improved in the revised version of the manuscript. However there are still several aspects that I hope the authors to improve as follows.
The authors moved former Fig. 5 and 6 as Supplementary Fig 1 and 2 in the revised version. I think these results are very important and should be included in the main body of the manuscript. I would like to recommend the authors that Sup Fig. 1 and 2 should be added as Fig. 3g and 4g, or Fig. 5a and 5b, respectively.
Response:: As was suggested by the reviewers the figures Sup Fig. 1 and 2 as Figures 5 and 5 b were incorporated into the main text of the manuscript.
Comment: It is clear that dynamic alteration of DNA methylation occurs with seed drying and aging. I would like to suggest to discuss (or to mention) whether these dynamic DNA methylation changes occur in embryonic tissues and/or endosperm.
Response: As for the analyses we used small Populus nigra seeds, DNA methylation changes were analysed in entire seeds. However, as we previously showed that the stress of desiccation can affect differently the methylation level in embryonic axes and cotyledons of recalcitrant and orthodox seeds of Acer sp., therefore for further research it would be interesting to compare the aging process’s impact on separate tissues from seeds where the isolation of both tissues is easy. We strongly appreciate this suggestion. We added a research clarifying statement in the conclusion section as follows: “ Even though the mechanisms behind DNA demethylation at hydrated and dry cytoplasm still need to be investigated, we were able to demonstrate that changes in the global level of 5mC analyzed in DNA isolated from entire poplar seeds precede a decline in seed viability.”
Comment: Minor points
Line 325. The R value should be corrected.
Response: We double checked all R values in the manuscript, and now they are correct.
Comment:Line 436. Tab. 1 should be Table 1.
Response: Corrected
Comment: Line 479. “base excision repair process (BER) process” should be “base excision repair (BER) process”.
Response: Corrected